# A guanosine tetraphosphate (ppGpp) mediated brake on photosynthesis is required for acclimation to nitrogen limitation in *Arabidopsis*

Shanna Romand[1], Hela Abdelkefi[1,2], Cécile Lecampion[1], Mohamed Belaroussi[1], Melanie Dussenne[1], Brigitte Ksas[3], Sylvie Citerne[4], Jose Caius[5,6], Stefano D'Alessandro[1], Hatem Fakhfakh[2,7], Stefano Caffarri[1], Michel Havaux[3], Ben Field[1]*

[1]Aix-Marseille University, CEA, CNRS, BIAM, LGBP Team, Marseille, France; [2]University of Tunis El Manar, Faculty of Sciences of Tunis, Laboratory of Molecular Genetics, Immunology and Biotechnology, Tunis, Tunisia; [3]Aix-Marseille University, CEA, CNRS, BIAM, SAVE Team, Saint-Paul-lez-Durance, France; [4]Institut Jean-Pierre Bourgin, UMR1318 INRA-AgroParisTech, INRAE Centre de Versailles-Grignon, Université Paris-Saclay, Versailles, France; [5]Université Paris-Saclay, CNRS, INRAE, Univ Evry, Institute of Plant Sciences Paris-Saclay (IPS2), Orsay, France; [6]Université de Paris, CNRS, INRAE, Institute of Plant Sciences Paris-Saclay (IPS2), Orsay, France; [7]University of Carthage, Faculty of Sciences of Bizerte, Bizerte, Tunisia

*For correspondence:
ben.field@univ-amu.fr

**Abstract** Guanosine pentaphosphate and tetraphosphate (together referred to as ppGpp) are hyperphosphorylated nucleotides found in bacteria and the chloroplasts of plants and algae. In plants and algae artificial ppGpp accumulation can inhibit chloroplast gene expression, and influence photosynthesis, nutrient remobilization, growth, and immunity. However, it is so far unknown whether ppGpp is required for abiotic stress acclimation in plants. Here, we demonstrate that ppGpp biosynthesis is necessary for acclimation to nitrogen starvation in *Arabidopsis*. We show that ppGpp is required for remodeling the photosynthetic electron transport chain to downregulate photosynthetic activity and for protection against oxidative stress. Furthermore, we demonstrate that ppGpp is required for coupling chloroplastic and nuclear gene expression during nitrogen starvation. Altogether, our work indicates that ppGpp is a pivotal regulator of chloroplast activity for stress acclimation in plants.

## Editor's evaluation

The authors provide genetic evidence showing that the unusual nucleotide, guanosine tetraphosphate (ppGpp), modulates plastidial gene expression and photosynthetic electron transfer upon nitrogen starvation in Arabidopsis. The results provide further insight into functional importance of the plastidial stringent response in plant physiology.

## Introduction

Plants cannot easily escape harsh environmental fluctuations, and so their survival hinges on facing each threat. To this end plants have developed intricate stress perception and response mechanisms

(*Devireddy et al., 2021*), where the chloroplast is recognized as both a major signaling hub and a target for acclimation (*Kleine et al., 2021*). Likely candidates for regulating chloroplast stress signaling are the hyperphosphorylated nucleotides guanosine pentaphosphate and tetraphosphate (together referred to as ppGpp) that are synthesized from ATP and GDP/GTP by chloroplast localized enzymes of the RelA SpoT Homologue (RSH) family (*Boniecka et al., 2017*; *Field, 2018*). In bacteria, where ppGpp was originally discovered, a considerable body of work indicates that ppGpp and related nucleotides interact directly with specific effector enzymes to regulate growth rate and promote stress acclimation (*Ronneau and Hallez, 2019*; *Bange et al., 2021*; *Anderson et al., 2021*). In plants, ppGpp signaling is less well understood both at the physiological and mechanistic levels. ppGpp levels increase transiently in diverse plants in response to treatment with a range of abiotic stresses and stress-related hormones (abscisic acid, jasmonate, and ethylene) (*Takahashi et al., 2004*; *Ihara et al., 2015*). The accumulation of ppGpp is controlled by the ppGpp synthesis and hydrolysis activity of three families of chloroplast targeted RSH enzyme, RSH1, RSH2/3, and RSH4/CRSH (*Atkinson et al., 2011*; *Ito et al., 2017*; *Avilan et al., 2019*). In the angiosperm *Arabidopsis thaliana* (*Arabidopsis*), where ppGpp signaling is best characterized, RSH1 lacks ppGpp synthase activity and acts as the main ppGpp hydrolase (*Sugliani et al., 2016*), the closely related RSH2 and RSH3 act as the major ppGpp synthases during the day (*Mizusawa et al., 2008*; *Maekawa et al., 2015*; *Sugliani et al., 2016*), and the calcium-activated RSH (CRSH) is responsible for ppGpp synthesis at night and in response to darkness (*Ihara et al., 2015*; *Ono et al., 2020*). RSH2 and RSH3 are bifunctional ppGpp synthase/hydrolase enzymes, and their hydrolase activity was recently shown to be necessary for constraining CRSH-mediated ppGpp production at night (*Ono et al., 2020*). The artificial accumulation of ppGpp itself has been shown to repress the expression of certain chloroplast genes in vivo via the inhibition of chloroplast transcription (*Yamburenko et al., 2015*; *Maekawa et al., 2015*; *Sugliani et al., 2016*; *Ono et al., 2020*). The chloroplastic effectors of ppGpp signaling in plants have not yet been identified. Nevertheless, ppGpp is able to specifically repress activity of the chloroplast guanylate kinase, suggesting the existence of a ppGpp effector mechanism that is found in many firmicute bacteria (*Nomura et al., 2014*; *Field, 2018*; *Bange et al., 2021*). According to the firmicute model, guanylate kinase inhibition causes a drop in GTP concentration, which in turn results in reduced transcription from rRNA-coding genes where GTP is usually the initiating nucleotide. Over accumulation of ppGpp in the chloroplast by the overexpression of RSH3 or bacterial ppGpp synthase domains has also shown that ppGpp may act as a conserved repressor of photosynthesis in both land plants (*Maekawa et al., 2015*; *Sugliani et al., 2016*; *Honoki et al., 2018*; *Harchouni et al., 2021*) and algae (*Avilan et al., 2021*). Interestingly, the artificial accumulation of ppGpp also appears to protect plants against nitrogen deprivation (*Maekawa et al., 2015*; *Honoki et al., 2018*). In addition, a few studies have now demonstrated that ppGpp is required for different physiological processes; these include the regulation of plant growth and development (*Sugliani et al., 2016*; *Ono et al., 2020*), plant immunity (*Abdelkefi et al., 2018*), and photosynthesis under standard growth conditions (*Sugliani et al., 2016*). However, there are so far no demonstrations that ppGpp is required for abiotic stress acclimation in plants.

Here, we looked at the role of ppGpp in acclimation to nitrogen deprivation. ppGpp was discovered in bacteria during research into the acclimation of *Escherichia coli* to amino acid limitation (*Cashel and Gallant, 1969*). We reasoned that a similar role was likely to be maintained in plants due to a shared and fundamental requirement for amino acids and the nitrogen to synthesize them. In particular, chloroplasts, and the photosynthetic machinery within, represent a major nitrogen store that must be remobilized during periods of nitrogen limitation and during senescence. We found that ppGpp biosynthesis by RSH2 and RSH3 is necessary for acclimation to nitrogen starvation in *Arabidopsis*. During nitrogen starvation ppGpp acts by remodeling the photosynthetic electron transport chain to downregulate photosynthetic activity, and by protecting against oxidative stress and tissue damage. Furthermore, we show that ppGpp couples chloroplastic and nuclear gene expression during nitrogen starvation. Overall, our work indicates that ppGpp is a pivotal regulator of chloroplast activity with a photoprotective role and that ppGpp signaling is required for the acclimation of plants to harsh environmental conditions.

## Results

### ppGpp is required for acclimation to nitrogen deprivation

To determine whether ppGpp plays a significant physiological role during nitrogen starvation we grew a series of *RSH* lines on a nitrogen limiting medium that imposes a progressive nitrogen starvation (*Figure 1A*). The *RSH* lines were previously shown to accumulate lower (OX:RSH1, *RSH* quadruple mutant [$rsh_{QM}$]) or higher (*rsh1-1*) amounts of ppGpp under standard growth conditions (*Sugliani et al., 2016*). Growth arrest occurred for all lines 9–10 days after sowing on nitrogen limiting media and was followed by the production of anthocyanins and loss of chlorophyll (*Figure 1A*, *Figure 1—figure supplement 1*). Strikingly, we observed that the low ppGpp lines OX:RSH1 and $rsh_{QM}$ showed significantly higher rates of cotyledon death than in the corresponding wild-type plants (*Figure 1A, B*). Interestingly, between 16 and 22 days we also saw a greater rate of new leaf initiation in the ppGpp-deficient lines (*Figure 1A*).

We reasoned that the increased cotyledon death in the low ppGpp lines could be due to overproduction of reactive-oxygen species (ROS). We therefore imaged the autoluminescence of lipid peroxides, a signature of ROS accumulation (*Figure 1C*). Under nitrogen deprivation conditions strong autoluminescence was observed only in the OX:RSH1 and $rsh_{QM}$ low ppGpp lines, while high ppGpp *rsh1-1* plants showed almost no autoluminescence (*Figure 1C*). Quantification of lipid peroxidation products (hydroxy-octadecatrienoic acids [HOTEs], the oxidation products of linolenic acid, the major fatty acid in *Arabidopsis* leaves) by HPLC further supported these findings: HOTEs increased in the wild type in response to nitrogen deprivation, and this increase was significantly larger in the ppGpp-deficient lines (*Figure 1D*). These results indicate that nitrogen deprivation promotes ROS accumulation, and that ppGpp is required to prevent overaccumulation of ROS, oxidative stress, and death of cotyledons under these conditions.

### ppGpp levels increase at an early stage of nitrogen deprivation

In bacteria and in plants, ppGpp is known to peak in response to stress before stabilizing at lower levels (*Varik et al., 2017*; *Takahashi et al., 2004*; *Ihara et al., 2015*). To determine the kinetics of changes in ppGpp concentration during nitrogen deprivation we quantified ppGpp levels in wild-type plants at different timepoints during growth on nitrogen limiting media. We observed a peak in ppGpp levels after 8 days on nitrogen limiting medium that was not observed in plants of the same age grown on nitrogen replete medium (*Figure 1E*). The ppGpp concentration decreased and stabilized after 12 days on nitrogen limiting medium. In contrast, the level of GTP decreased throughout the timecourse. We next determined how (p)ppGpp levels were affected in each of the *RSH* lines during growth on nitrogen limiting medium (*Figure 1F*). As anticipated we found that ppGpp levels in OX:RSH1 and $rsh_{QM}$ lines were lower than in wild-type plants, while they were higher in *rsh1-1* plants. Strikingly, we found that the lines with low ppGpp levels (OX:RSH1 and $rsh_{QM}$) maintained higher levels of GTP than in the wild-type controls or *rsh1-1* (*Figure 1F*), resulting in dramatic differences in the ppGpp/GTP ratio (*Figure 1—figure supplement 2*). Therefore, nitrogen limitation leads to an early increase in ppGpp levels and is followed by a progressive diminution of the GTP pool that is dependent on ppGpp accumulation. These results suggest that ppGpp inhibits GTP biosynthesis, perhaps through inhibition of guanylate kinase (*Nomura et al., 2014*), and supports the hypothesis that plants have a ppGpp effector mechanism that operates through purine metabolism as found in many bacteria.

### Nitrogen deprivation promotes a ppGpp-dependent drop in photosynthetic activity

Under standard growth conditions artificial production of ppGpp has been shown to downregulate photosynthetic activity in *Arabidopsis* (*Sugliani et al., 2016*; *Maekawa et al., 2015*). Defective downregulation of photosynthesis during nitrogen starvation could cause the overaccumulation of ROS and lipid peroxides observed in the low ppGpp lines due to oversaturation of photosynthesis in the presence of diminished metabolic sinks. We therefore measured different photosynthetic parameters in the wild type and *RSH* lines during growth on nitrogen limiting medium. A decrease in the maximum quantum yield (Fv/Fm) of photosystem II (PSII) caused by an increase in basal fluorescence (Fo, see Materials and methods for formula) was observed in the cotyledons and first set of true leaves

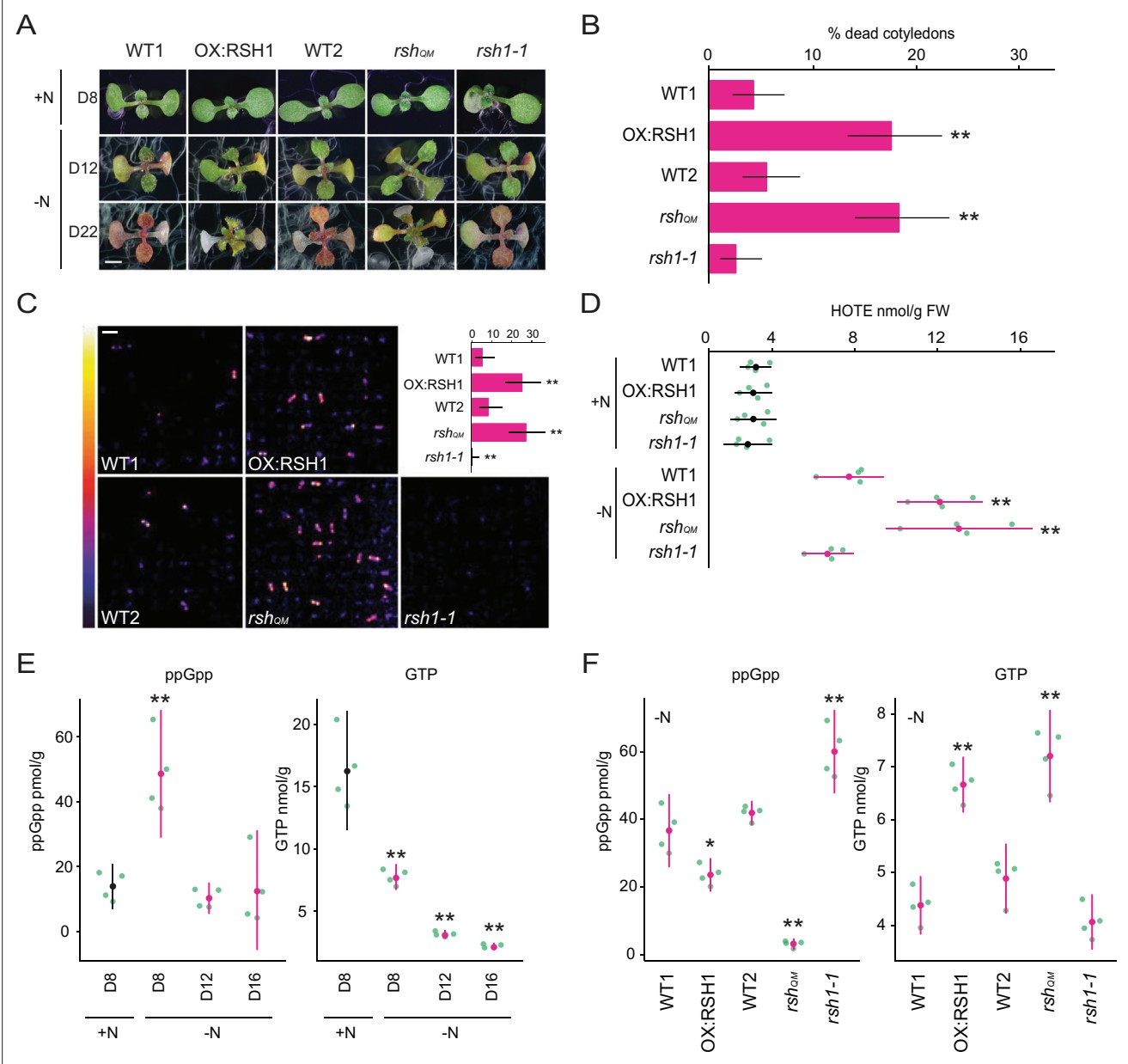

**Figure 1.** ppGpp is required for acclimation to nitrogen deprivation. (**A**) Images of seedlings grown on nitrogen replete (+N) or nitrogen limiting (−N) medium for 8 (D8), 12 (D12), and 22 (D22) days. Scale, 3 mm. WT1 (Col-0) and WT2 (*qrt1-2*) are the wild type for OX:RSH1 and *rsh$_{QM}$/rsh1-1*, respectively. See *Figure 1—figure supplement 1* for additional images. (**B**) Percentage of plants with dead cotyledons (completely white with collapsed tissue) for different genotypes grown on −N medium for 22 days. Three pooled experimental replicates, median ± 95% confidence interval (CI), n = 285–298 seedlings per genotype. (**C**) Bioluminescence emission from lipid peroxides in seedlings grown on −N medium for 16 days. Inset graph shows median number of luminescent seedlings ± 95% CI, n = 100 seedlings. Scale, 1.1 cm. (**D**) Quantification of hydroxy-octadecatrienoic acids (HOTEs) from seedlings grown in −N for 12 days or +N for 8 days where the developmental stage is similar. Mean ± 95% CI, n = 4 experimental replicates. Concentrations of ppGpp and GTP in wild-type plants were determined (**E**) at the indicated time points during growth on +N and −N medium and (**F**) in different genotypes after 12 days of growth in −N medium (equivalent to 10 days in the experiment in panel E). The ppGpp/GTP ratio is shown in *Figure 1—figure supplement 2*. Mean ± 95% CI, n = 4 experimental replicates. Statistical tests shown against respective wild-type controls, *p < 0.05, **p < 0.01. Source data and statistical test reports are shown in *Figure 1—source data 1*.

The online version of this article includes the following source data and figure supplement(s) for figure 1:

**Source data 1.** Source data and statistical test reports for *Figure 1* and supplements.

**Figure supplement 1.** Extended timecourse of nitrogen deprivation.

**Figure supplement 2.** GTP/ppGpp ratios in RSH lines.

of wild-type seedlings (*Figure 2A, B*, *Figure 2—figure supplement 1*, *Figure 2—source data 1*). Remarkably, this decrease was almost completely suppressed in the low ppGpp *rsh*<sub>QM</sub> and OX:RSH1 lines, and was enhanced in the high ppGpp *rsh1-1* mutant (*Figure 2A, B*, *Figure 2—figure supplement 2A*). Analysis of *rsh2-1* and *rsh3-1* single mutants, as well as the complementation of an *rsh2-1 rsh3-1* double mutant with the genomic version of *RSH3*, indicates that the RSH3 enzyme is the major ppGpp synthase responsible for driving the Fv/Fm decrease under nitrogen starvation conditions (*Figure 2—figure supplement 2B*). Likewise, the enhanced Fv/Fm decrease in the high ppGpp *rsh1-1* mutant and complementation of the *rsh1-1* phenotype by expression of the genomic *RSH1* indicate that the ppGpp hydrolase RSH1 acts antagonistically to RSH3, probably by constraining ppGpp accumulation during nitrogen deprivation (*Figures 1F and 2A, B*, *Figure 2—figure supplement 2C*).

While unlikely, it is possible that RSH enzymes could influence Fv/Fm via mechanisms that do not require ppGpp synthesis or hydrolysis. To exclude this possibility we conditionally overexpressed a heterologous chloroplast targeted ppGpp hydrolase from *Drosophila melanogaster* (Metazoan SpoT homolog, MESH)(*Sugliani et al., 2016 Figure 2—figure supplement 2D*). We observed that ppGpp depletion by MESH suppresses the Fv/Fm decrease in response to nitrogen limitation, confirming a specific role for ppGpp in this process.

Our experiments were carried out on plate grown seedlings and in the presence of exogenous sugar in the media. To test the robustness of our observations to different experimental setups we exposed mature plants grown in quartz sand to nitrogen starvation conditions (*Figure 2—figure supplement 2E*). We observed responses from the wild type, *rsh*<sub>QM</sub> and *rsh1-1* mutant that were consistent with those observed for plate grown seedlings, confirming the robustness of the ppGpp-dependent decrease in Fv/Fm in response to nitrogen limitation.

Fv/Fm measurements provide information principally on PSII. To understand the state of the entire photosynthetic electron transport chain we next measured the relative rate of electron transport (ETR). We found that nitrogen deprivation led to a large decrease in ETR in wild-type plants (*Figure 2C*, *Figure 2—figure supplement 4A*). The *RSH* lines showed similar ETRs to the wild type under nitrogen replete conditions. However, nitrogen deprivation led to large differences in the *RSH* lines compared to the wild type: a low ETR was observed in the high ppGpp *rsh1-1* mutant, and a substantially higher ETR in the low ppGpp *rsh*<sub>QM</sub> and OX:RSH1 lines. Induction of the MESH ppGpp hydrolase also prevented much of the decrease in ETR observed under nitrogen deprivation, again indicating that this phenomenon is dependent on the activity of RSH enzymes and ppGpp accumulation (*Figure 2—figure supplement 4B*).

## The ppGpp-dependent decline in photosynthesis occurs even at low light fluences

The ppGpp-dependent drop in Fv/Fm during nitrogen deprivation is reminiscent of photoinhibition, a process whose rate is proportional to light intensity (*Tyystjärvi and Aro, 1996*). We therefore asked whether the ppGpp-dependent decrease in Fv/Fm is similarly dependent on the excitation pressure on the photosystems. Strikingly, under nitrogen deprivation the Fv/Fm of the wild type decreased to a similar minimum regardless of the light intensity (photosynthetic photon flux density; low light, 10 μmol m$^{-2}$ s$^{-1}$; growth light, 80 μmol m$^{-2}$ s$^{-1}$; and higher light, 150 μmol m$^{-2}$ s$^{-1}$)(*Figure 2D*). However, the rate of the Fv/Fm decrease was proportional to light intensity, showing a slower decrease to the minimum under low light than under normal light or high light. A similar phenomenon was observed in the high ppGpp *rsh1-1* line, except that the Fv/Fm dropped at a faster rate and to a considerably lower level. In contrast, in the low ppGpp *rsh*<sub>QM</sub> mutant we observed almost no decrease in Fv/Fm, regardless of the light intensity tested. The insensitivity of the *rsh*<sub>QM</sub> mutant indicates that the process is completely dependent on ppGpp biosynthesis. Furthermore, the response of the wild-type and even stronger response of the *rsh1-1* mutant at all light fluences indicate that ppGpp-initiated regulation of Fv/Fm is triggered in response to nitrogen deprivation rather than to changes in the excitation status of the photosynthetic electron transport chain (*Figure 2D*).

## ppGpp is required for the timely degradation of photosynthetic proteins

In order to better understand the changes in the photosynthetic machinery that underlie the reduction of photosynthetic capacity during nitrogen starvation we analyzed the abundance of representative

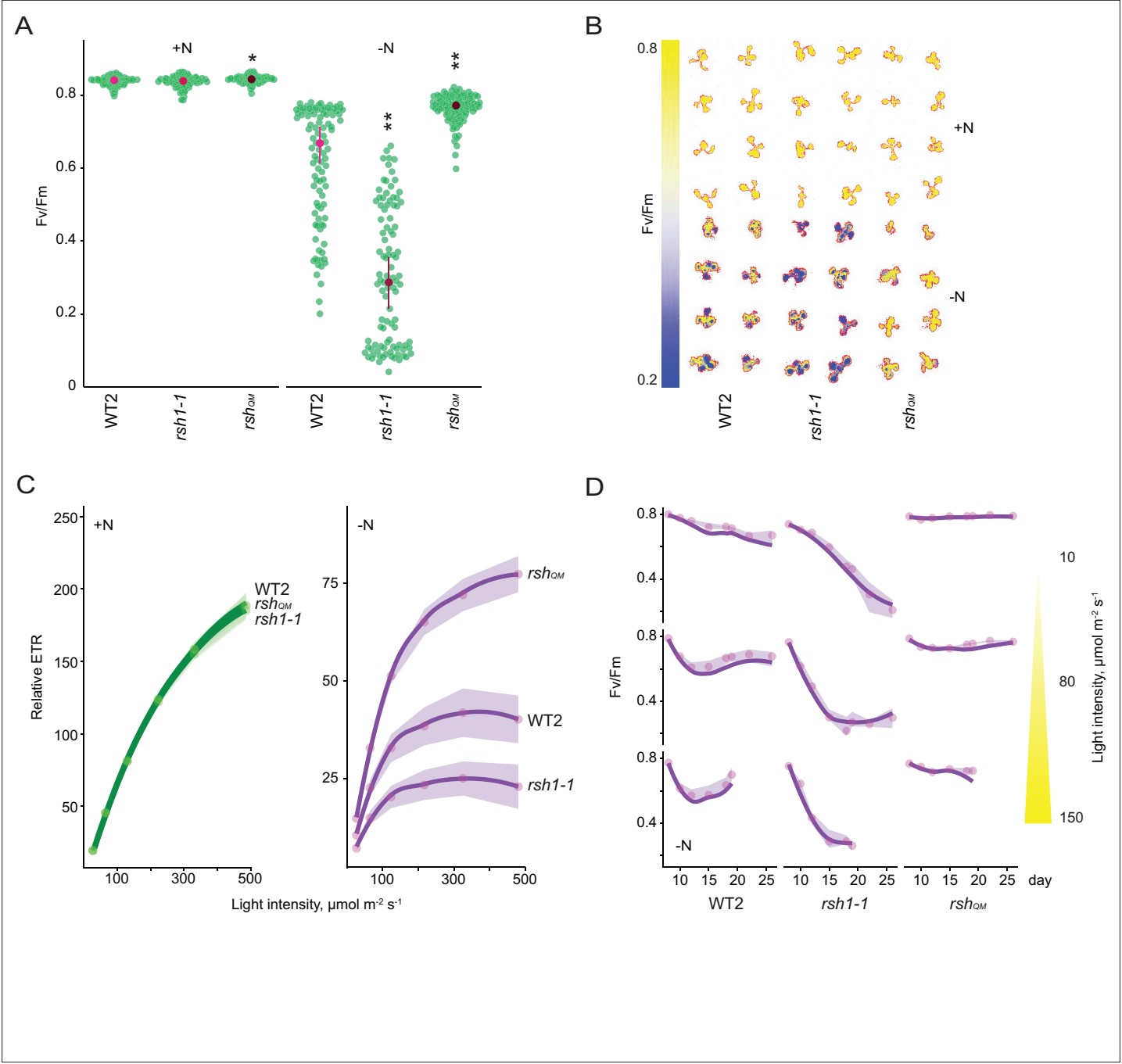

**Figure 2.** Nitrogen deprivation promotes a ppGpp-dependent drop in photosynthetic capacity. Seedlings were grown 8 days on nitrogen replete media (+N) or 12 days on nitrogen limiting (−N) media and (**A**) the maximal yield of PSII (Fv/Fm) measured by fluorescence imaging individual seedlings. Median ± 95% confidence interval (CI), $n$ = 95–100 seedlings. (**B**) Fv/Fm images of whole seedlings grown on +N and −N media for 12 days. (**C**) Relative electron transport rate (ETR) measurements in different lines grown 8 days on +N media or 12 days on −N media. Median ± 95% CI, $n$ = 95–100 seedlings. (**D**) Fv/Fm timecourses from seedlings grown on −N media and transferred to three different light intensities (photosynthetic photon flux density, 10, 80, and 150 µmol m$^{-2}$ s$^{-1}$) after 6 days. Median ± 95% CI, $n$ = 95–100 seedlings. Tests shown against respective wild-type controls, *p < 0.05, **p < 0.01. Additional supporting data are presented in *Figure 2—figure supplement 2*, *Figure 2—figure supplement 3* and *Figure 2—figure supplement 4*. Source data and statistical test reports are shown in *Figure 2—source data 1*.

The online version of this article includes the following source data and figure supplement(s) for figure 2:

**Source data 1.** Source data and statistical test reports for *Figure 2* and supplements.

**Figure supplement 1.** Changes in additional photosynthetic parameters (Fo, Fm, and Fo/Fm) for lines shown in panel 2A.

*Figure 2 continued on next page*

*Figure 2 continued*

**Figure supplement 2.** Role of ppGpp in the nitrogen deprivation induced decrease in PSII maximal yield.

**Figure supplement 3.** Changes in additional photosynthetic parameters (Fo, Fm, and Fo/Fm) for *Figure 2—figure supplement 2*.

**Figure supplement 4.** ppGpp is required for reducing electron transport rate (ETR) during nitrogen deprivation.

photosynthetic proteins in the wild type and the low ppGpp line OX:RSH1 (*Figure 3A*). In the wild type, we observed a decrease in proteins representative of nearly all the major photosynthetic complexes analyzed (PsbA, Lhcb1, PsbO, PetA, PsaD, Lhca1, and RBCL) and PTOX analyzed between 8 and 16 days of growth under nitrogen deprivation. The decrease in photosynthetic proteins was defective in OX:RSH1 plants, which showed a marked delay for the decrease in abundance of PsbA, PsbO, PetA,

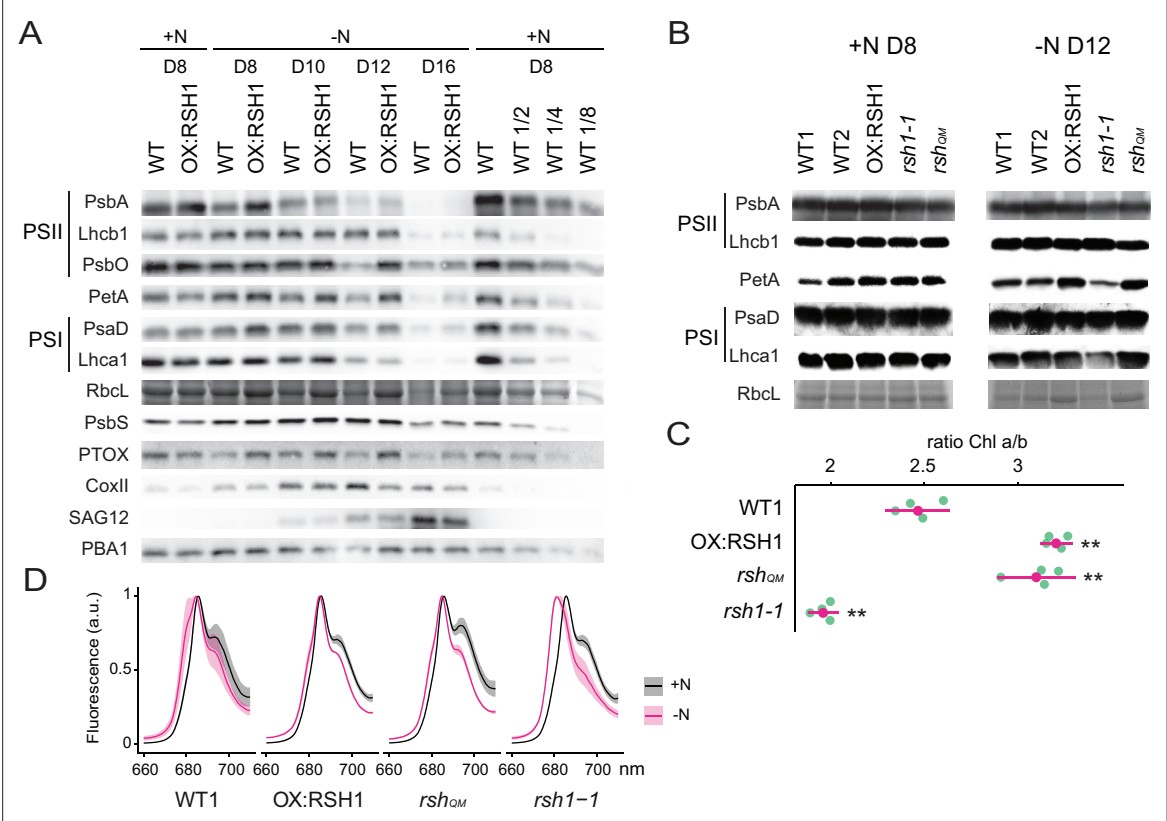

**Figure 3.** ppGpp-dependent alterations in the photosynthetic machinery during nitrogen deficiency. (**A**) Immunoblots showing evolution in abundance of the indicated proteins in seedlings grown in nitrogen replete (+N) or nitrogen limiting (−N) media for the indicated number of days. RbcL was revealed by Coomassie Brilliant Blue. Equal quantities of total proteins were loaded and PBA1, a subunit of the proteasome, was used as a protein normalization control (**B**). Immunoblots showing the abundance of the indicated proteins in purified thylakoid membranes from seedlings grown in +N for 8 days or −N for 12 days. RbcL was revealed by Coomassie Brilliant Blue staining. Equal quantities of total chlorophyll were loaded. Immunoblots after ppGpp depletion by induction of chloroplastic MESH are shown in *Figure 3—figure supplement 1*. (**C**) Chlorophyll a/b ratios in extracts from seedlings subjected to −N for 12 days. Means ± 95% confidence interval (CI), data from four experimental replicates. (**D**) Emission spectrum of chlorophyll fluorescence at 77 K between 660 and 720 nm, normalized to the PSII peak at 685 nm. The full spectra are shown in *Figure 2—figure supplement 2*. Measurements were made on seedlings grown in +N for 8 days or −N for 12 days. Means ± 95% CI; data from four experimental replicates. Statistical tests, **p < 0.01. Uncropped immunoblots are available in *Figure 3—source data 1* for 3A and *Figure 3—source data 2* for 3B. Numeric data and statistical test reports are shown in *Figure 3—source data 3*.

The online version of this article includes the following source data and figure supplement(s) for figure 3:

**Source data 1.** Uncropped immunoblots for 3A.

**Source data 2.** Uncropped immunoblots for 3B.

**Source data 3.** Source data and statistical test reports for *Figure 3* and supplements.

**Figure supplement 1.** ppGpp depletion by MESH also affects abundance of chloroplast proteins.

**Figure supplement 2.** Full 77°K chlorophyll fluorescence spectra under nitrogen deprivation.

PTOX, PsaD, and RBCL. This delay was especially visible after 12 days on nitrogen limiting medium. On the contrary, changes in Lhcb1 and Lhca1, the nuclear-encoded light harvesting proteins of PSII and PSI, were indistinguishable between wild-type and OX:RSH1 plants.

In contrast to the downregulation of many components of the photosynthetic chain during nitrogen deprivation, mitochondrial activity appeared to increase. The mitochondrial marker protein COXII, a subunit of the mitochondrial complex IV, increased to reach a maximum on day 12 of nitrogen deprivation. Interestingly, the COXII maximum was higher in the wild type than in OX:RSH1 plants suggesting increased ppGpp levels could also positively influence mitochondrial activity. PsbS, a key component of nonphotochemical quenching, and SAG12, a marker of senescence, showed similar patterns of accumulation in WT and OX:RSH1 plants.

We next extended our analysis from OX:RSH1 to the other RSH lines to confirm that ppGpp is responsible for the observed changes in photosynthetic proteins during nitrogen deprivation (*Figure 3B*). In this experiment we normalized to total chlorophyll to exclude possible artifacts linked to normalization on total protein, yet we obtained broadly similar results. PetA and RBCL, the proteins with the greatest differences in OX:RSH1 during the timecourse, were more abundant in the low ppGpp lines OX:RSH1 and $rsh_{QM}$ at day 12 under nitrogen deprivation. The same proteins were also less abundant than the wild type in the high ppGpp line *rsh1-1*. Interestingly, PsbA accumulation did not appear different between the lines, contrasting with results from the OX:RSH1 timecourse. However, this difference is likely linked to the chlorophyll normalization because Lhcb1 levels were higher in the wild type and *rsh1-1* mutant, indicating a similar direction of change for the ratio of PSII core to antenna. Finally, to unequivocally link these changes to ppGpp we analyzed representative protein levels in induced MESH ppGpp hydrolase lines, which again showed the ppGpp is required for diminution of PetA and RBCL levels during nitrogen deprivation (*Figure 3—figure supplement 1*). Altogether our immunoblotting experiments indicate the ppGpp is required for the timely reduction in abundance of key photosynthetic proteins during nitrogen starvation, and that proteins such as PsbO (PSII core), PetA (Cyt b₆f), and RBCL (Rubisco) appear to be preferentially affected.

## Nitrogen starvation promotes ppGpp-dependent energetic uncoupling of PSII antenna

The ppGpp-dependent drop in Fv/Fm and drop in the PsbA/Lhcb1 ratio suggest that ppGpp specifically remodels the structure of the PSII complex during nitrogen deprivation. Supporting this idea, we also found a ppGpp-dependent drop in the Chl a/b ratio (*Figure 3C*), indicating a substantial decrease in PSII RC, which lacks Chl b, to Chl b-rich PSII antenna. An increase in relative PSII antenna abundance might be accompanied by the energetic uncoupling from the PSII RC. Low temperature chlorophyll fluorescence spectra confirmed this hypothesis. During nitrogen deprivation there was a shift of the PSII antenna emission peak toward lower wavelengths in the wild type, indicating an energetic decoupling from PSII RC (*Crepin et al., 2016*; *Figure 3D*, *Figure 3—figure supplement 2*). This shift is mainly ppGpp dependent because it was smaller in the low ppGpp OX:RSH1 and $rsh_{QM}$ lines, and much stronger in the high ppGpp *rsh1-1* line. Therefore, multiple approaches show that ppGpp is necessary for PSII remodeling and decreasing the excitation pressure on the photosystems during nitrogen starvation.

## ppGpp plays a major role during acclimation to nitrogen deprivation

Our results show that defects in the capacity of plants to accumulate ppGpp lead to aberrant photosynthesis and stress phenotypes under nitrogen deprivation. To determine the extent and specificity of the impact of ppGpp on cellular processes we analyzed global nuclear and organellar transcript abundance in wild-type and OX:RSH1 plants.

Nitrogen deprivation caused a massive alteration in transcript levels in both the wild type and OX:RSH1, significantly affecting the accumulation of about 15,000 transcripts (*Figure 4A*, *Figure 4—source data 1*). For the wild type, there was substantial overlap (80%) with differentially accumulating transcripts recently identified in plants grown under chronic nitrogen limitation for 60 days (*Luo et al., 2020*). Strikingly, we observed large increases in *RSH2* and *RSH3* transcript levels in response to nitrogen deprivation (*Figure 4B*). These were accompanied by a decrease in *RSH1* transcript levels, suggesting a coordinated upregulation of ppGpp biosynthetic capacity. The expression profile of OX:RSH1 plants showed clear differences to the wild type, especially under nitrogen deprivation.

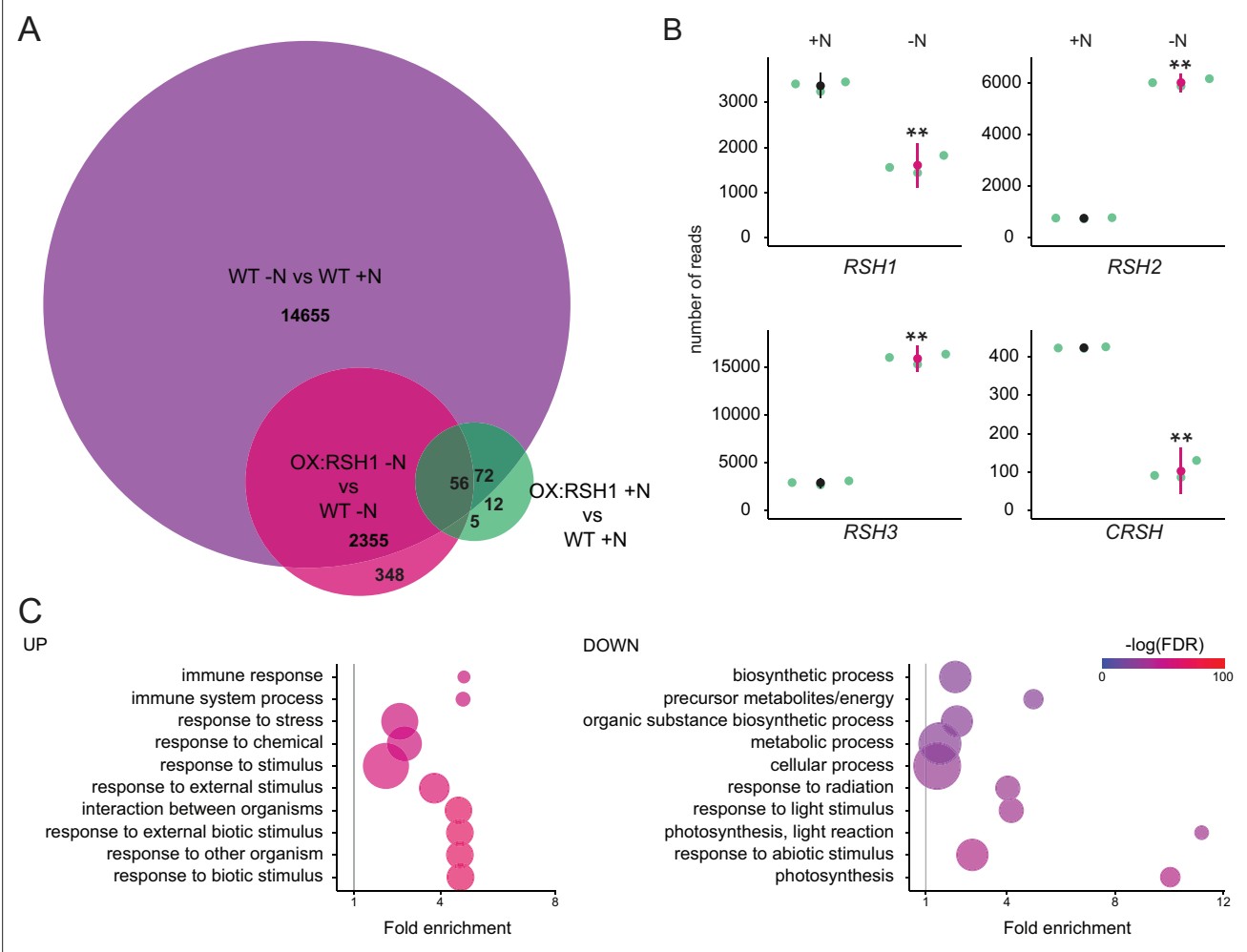

**Figure 4.** ppGpp plays a major role during acclimation to nitrogen deprivation. RNA-seq experiments were performed on WT and OX:RSH1 seedlings grown 8 days on nitrogen replete media (+N) or 12 days on nitrogen limiting (−N) media, n = 3 experimental replicates. (**A**) Venn diagram for transcripts showing differential accumulation for each of three comparisons. All differentially accumulating transcripts are listed in *Figure 4—source data 1*. (**B**) RNA-seq transcript levels for the four *RSH* genes in the WT, +N vs −N. Mean ± 95% confidence interval (CI), **p < 0.01. (**C**) Enriched gene ontology terms among significantly up- and downregulated transcripts in OX:RSH1 vs WT under −N. The 10 most significant terms are shown, point size is proportional to gene number. FDR, false discovery rate. The full GO analysis is presented in *Figure 4—source data 2*. Source data and statistical test reports are shown in *Figure 4—source data 3*.

The online version of this article includes the following source data for figure 4:

**Source data 1.** Differentially accumulating transcripts.

**Source data 2.** GO enrichment analysis.

**Source data 3.** Source data and statistical test reports for *Figure 4B*.

Under control conditions, OX:RSH1 plants had a very similar gene expression profile to the wild type, with the differential accumulation of only 150 transcripts (*Figure 4A*, *Figure 4—source data 1*). However, under nitrogen deprivation we observed the differential accumulation of 2700 transcripts in OX:RSH1. The greater deregulation of the OX:RSH1 transcript profile under nitrogen deprivation indicates that ppGpp plays a major and specific role in the acclimation response.

Analysis of enrichment for gene ontology terms corroborated our findings that low ppGpp plants are unable to properly acclimate to nitrogen deprivation. OX:RSH1 plants accumulated more transcripts associated with stress, including oxidative stress than the wild type (*Figure 4C*, *Figure 4—source data 2*). We also observed a significant decrease in the abundance of transcripts for genes involved in photosynthesis and chloroplast activity. Notably, these downregulated genes were all nucleus encoded (*Figure 4C*).

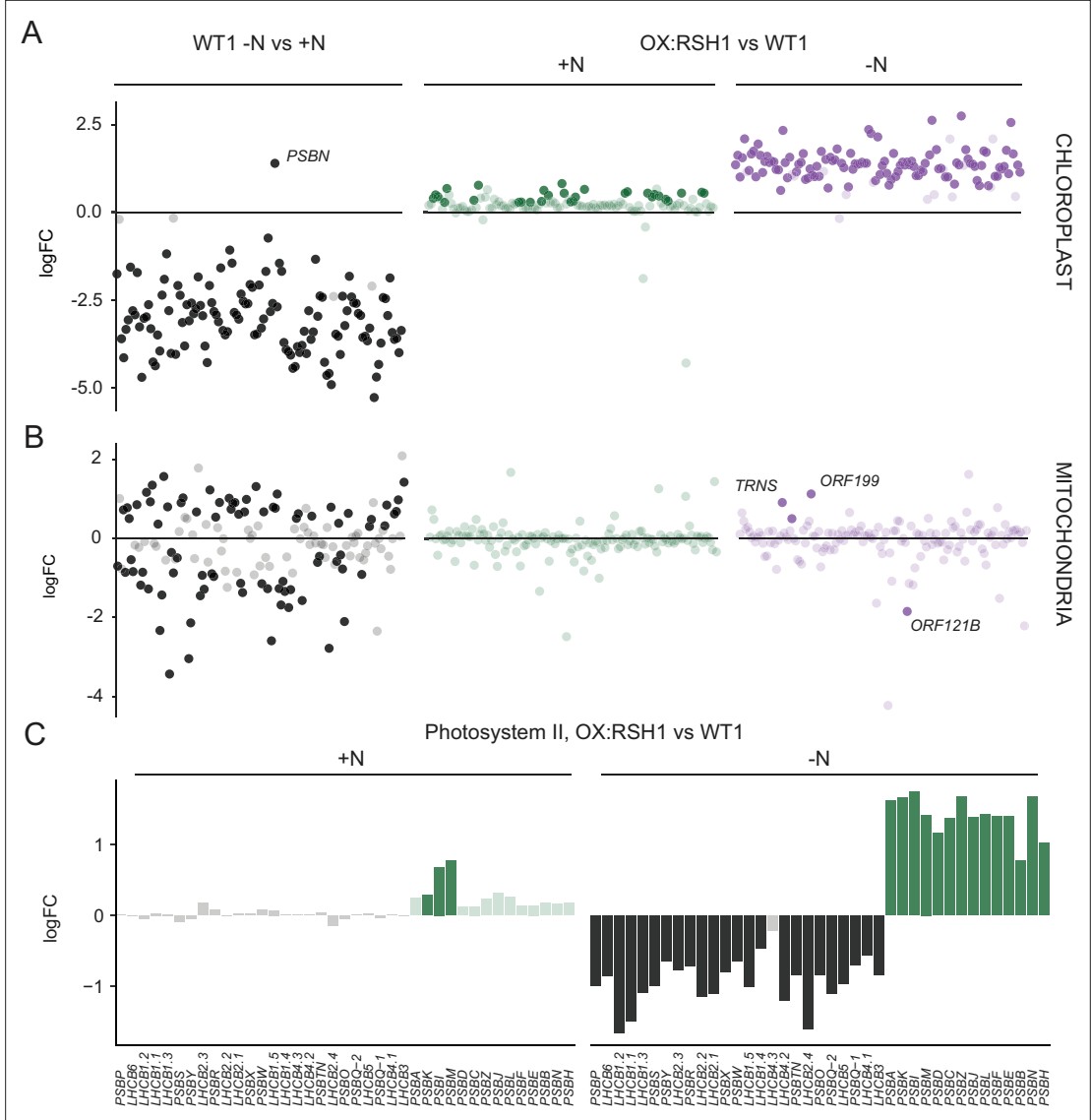

**Figure 5.** ppGpp is required for the downregulation of chloroplast gene expression during nitrogen deficiency. (**A**) The differential expression of chloroplast transcripts ordered along the chloroplast genome (*Figure 5—source data 1*). (**B**) The differential expression of mitochondrial transcripts ordered along the mitochondrial genome (*Figure 5—source data 2*). (**C**) Relative transcript levels in OX:RSH1 vs the wild-type control for nuclear (black) and chloroplast (green) genes encoding subunits of the photosystem II complex. Solid colors indicate significantly different changes in expression (p < 0.05), transparent colors indicate nonsignificant changes. Source data for 5C are shown in *Figure 5—source data 3*.

The online version of this article includes the following source data for figure 5:

**Source data 1.** Chloroplast gene expression.

**Source data 2.** Mitochondrial gene expression.

**Source data 3.** Source data for *Figure 5C* including extended figures for multiple photosynthetic complexes.

## ppGpp is required for global downregulation of chloroplast gene expression

We next turned to the effects of nitrogen deprivation on organellar gene expression. Nitrogen deprivation leads to a strong decrease in the levels of all chloroplast transcripts except *PsbN* in the wild type (*Figure 5A*, *Figure 5—source data 1*). The OX:RSH1 mutant showed minor differences to the wild type under normal growth conditions. However, under nitrogen starvation the vast majority of chloroplast transcripts (120 of 133 transcripts analyzed) were present at significantly higher levels in OX:RSH1 than in the wild type, indicating that ppGpp is required for globally reducing chloroplast

gene expression. In contrast to the monodirectional response of the chloroplast to nitrogen deprivation, we found that mitochondrial transcripts showed diverse responses in the wild type (*Figure 5B*, *Figure 5—source data 2*). Significantly, mitochondrial transcript levels in the OX:RSH1 line were highly similar to the wild type under control and nitrogen deprivation conditions. Taken together, these results indicate that ppGpp is required for the downregulation of chloroplast gene expression during nitrogen starvation and demonstrate that ppGpp acts specifically within the chloroplast.

## ppGpp is required for coordinating chloroplast and nuclear gene expression

Our gene expression analysis showed the ppGpp depletion during nitrogen starvation appeared to have opposite effects on transcript abundance for nuclear-encoded chloroplast genes and chloroplast-encoded genes. Many chloroplast protein complexes contain subunits encoded by both the nuclear and chloroplast genomes. Notably, in the wild-type chloroplast protein complexes showed coordinate downregulation of both nuclear and chloroplast transcripts in response to nitrogen deprivation (*Figure 5—source data 1*). However, ppGpp depletion caused widespread misregulation of transcript abundance for complex subunits (*Figure 5C*, *Figure 5—source data 3*). For the PSII complex, there was little difference between OX:RSH1 and the wild type under nitrogen replete conditions. However, in OX:RSH1 nitrogen deprivation caused the large-scale uncoupling of the coordination of gene expression between the nuclear and chloroplast genomes: all but one of the transcripts for nuclear-encoded subunits of PSII were downregulated, while all the transcripts for chloroplast-encoded subunits of PSII were upregulated (*Figure 5C*). Similar genome uncoupling was observed in OX:RSH1 for transcripts encoding subunits of the chloroplastic Cyt $b_6$f, PSI, ATP synthase, NDH, TIC, transcription and translation complexes (*Figure 5—source data 3*). These results indicate first that nitrogen deprivation promotes the coordinated downregulation of genes encoding chloroplastic proteins in the nuclear and chloroplastic genomes, and second that ppGpp plays a pivotal role in coupling gene expression between nuclear and chloroplast genomes under nitrogen limitation.

## Discussion

Nitrogen deprivation has long been known to cause a drop in photosynthetic capacity in plants (*Terashima and Evans, 1988*; *Nunes et al., 1993*; *Verhoeven et al., 1997*; *Lu and Zhang, 2000*; *Garai and Tripathy, 2017*). Here, we show that ppGpp signaling plays a major role in this process. We demonstrate that nitrogen starvation leads to an early and transient increase in ppGpp levels (*Figure 1E*), and that the capacity to synthesize and accumulate ppGpp is then required for protecting plants against excess ROS accumulation, tissue damage and stress (*Figures 1A–D and 4C*). We show that ppGpp is likely to mediate the acclimation response by affecting the ppGpp/GTP ratio (*Figure 1—figure supplement 2*), promoting the downregulation of photosynthetic capacity (*Figures 2A–D, 4C, and 5A*) and remodeling the photosynthetic machinery (*Figures 3A–D and 5C*). Finally, we show that ppGpp may function by specifically downregulating chloroplast transcript abundance during nitrogen deprivation to maintain an equilibrium between chloroplast and nuclear gene expression (*Figure 5A–C*, *Figure 5—source data 1*). ppGpp was initially discovered thanks to the identification of bacterial mutants that were unable to downregulate rRNA transcription during amino acid starvation (*Stent and Brenner, 1961*; *Cashel and Gallant, 1969*). Our results here therefore indicate that the fundamental involvement of ppGpp signaling in acclimation to nitrogen deprivation is likely to have been maintained despite the large evolutionary distance separating bacteria from plants that includes major shifts including the domestication of the cyanobacterial ancestor of the chloroplast, and the different regulatory logic and signaling networks of photosynthetic eukaryotes.

### The physiological relevance of ppGpp signaling in plants

Depletion or removal of ppGpp is usually necessary for directly establishing the implication of ppGpp in a physiological process. Up to now a handful of studies in plants have directly implicated ppGpp physiological processes by showing a requirement for ppGpp synthesis (*Sugliani et al., 2016*; *Abdelkefi et al., 2018*; *Honoki et al., 2018*). We now add to these studies by using multiple approaches, including three different ppGpp depletion methods (OX:RSH1, $rsh_{QM}$ and MESH), to demonstrate that ppGpp is required for acclimation to nitrogen deprivation. Previous work hinted that ppGpp may

play such a role by showing that plants ectopically overaccumulating ppGpp appear able to better withstand transfer to media lacking nitrogen (*Maekawa et al., 2015*; *Honoki et al., 2018*). Notably, *Honoki et al., 2018* also used a similar *rsh2 rsh3* double mutant in their experiments though, apart from a delay in Rubisco degradation, did not observe clear-cut differences from the wild type in terms of photosynthetic activity after transfer to media lacking nitrogen. The reasons for this are not clear, but it could be related to differences in experimental set up such as the timepoints for analysis, how nitrogen starvation was imposed, or the different RSH3 mutant allele used.

Our findings further reinforce the evidence that RSH enzymes function in the chloroplast. GFP fusion experiments have shown that RSH enzymes are localized to the chloroplasts (*Maekawa et al., 2015*), and the pleiotropic phenotypes of RSH3 overexpressing plants can be suppressed by chloroplast expression of a ppGpp hydrolase (*Sugliani et al., 2016*). However, some proteins, such as ATP synthase subunit beta-3, show low levels of dual targeting that are not detectable using full-length fluorescent protein fusions (*Sharma et al., 2019*). Coverage of the full mitochondrial transcriptome here shows that overexpression of RSH1 has almost no effect on the large-scale changes in mitochondrial transcript abundance that occur in response to nitrogen deprivation (*Figure 5B*). These results therefore confirm that RSH enzymes operate within the chloroplast.

## New insights into ppGpp signaling in planta

The kinetics of changes in ppGpp concentration during nitrogen deprivation provide new information on how ppGpp acts in planta. We observed a transient peak in ppGpp levels after 8 days of growth on nitrogen limiting medium (*Figure 1E*). Similar transient peaks of ppGpp accumulation were also observed in plants subjected to different stresses (*Takahashi et al., 2004*; *Ihara et al., 2015*; *Ono et al., 2020*), and are observed in bacterial cells (*Varik et al., 2017*). Notably, the peak in ppGpp that we observed in response to nitrogen deprivation occurs prior to major changes in Fv/Fm (*Figure 2D*), protein levels (*Figure 3A*), or anthocyanin accumulation (*Figure 1—figure supplement 1*). These factors along with the progressive nature of the nitrogen starvation suggest that a signaling mechanism activates ppGpp biosynthesis relatively soon after perception of nitrogen supply limitation. The upregulation of *RSH2* and *RSH3* expression (*Figure 4B*) indicates that there is at least a transcriptional component to this response. However, additional layers of regulation are likely, including for example the allosteric activation of RSH enzymes by interaction with other proteins or small molecules as found for bacterial RSH (*Ronneau and Hallez, 2019*; *Irving and Corrigan, 2018*). After the initial peak the ppGpp concentration then drops to levels not very different to those observed in plants grown in nitrogen replete conditions (*Figure 1E*). Furthermore, OX:RSH1 plants only show slightly lower levels of ppGpp than the wild type despite showing very similar phenotypes to the $rsh_{QM}$ mutant (*Figure 1F*). Together these observations indicate that ppGpp continues to exert an effect after peaking, suggesting the activation of irreversible processes or increased sensitivity to the presence of ppGpp. The second of these possibilities is supported by the phenotype of the high ppGpp *rsh1-1* mutant which displays much stronger photosynthesis phenotypes than wild-type plants despite a relatively modest increase in ppGpp levels (*Figures 1F–3A–D*).

GTP is a key nucleotide in ppGpp signaling because it is both the substrate for ppGpp biosynthesis and enzymes in its biosynthetic pathway such as guanylate kinase are potential ppGpp effectors (*Field, 2018*). We observed a striking decrease in GTP levels during nitrogen deprivation (*Figure 1E*). Remarkably GTP levels were closely and inversely correlated to ppGpp levels in the wild type and the *RSH* lines (*Figure 1E, F*). GTP is present at levels almost three orders of magnitude higher than ppGpp, so the relationship between ppGpp and GTP cannot be simply explained by the consumption of GTP for ppGpp biosynthesis. Therefore, ppGpp accumulation drives the depletion of the cellular GTP pools by other means, which could for example include the inhibition of enzymes involved in chloroplastic GTP biosynthesis such as guanylate kinase (*Nomura et al., 2014*). Even more remarkable is that the ppGpp-driven GTP depletion occurs specifically under nitrogen deprivation because accumulation of ppGpp to high levels in OX:RSH3 plants does not affect the GTP pool under normal growth conditions (*Bartoli et al., 2020*). The reduction in GTP levels may contribute to ppGpp signaling in two ways. First, reduced GTP levels are likely to directly contribute to the observed ppGpp-dependent reduction in chloroplast transcription in a similar fashion to ppGpp signaling in firmicute bacteria (*Bange et al., 2021*). Second, the increase in the ppGpp/GTP ratio will significantly augment the action of ppGpp on effectors where ppGpp competes with GTP for binding. GTP-binding effectors

are common in bacteria, and many are conserved in chloroplasts, particularly those involved in translation and ribosome biogenesis (*Pausch et al., 2018*; *Field, 2018*). We propose that the increase in the ppGpp/GTP ratio during nitrogen limitation is may also explain how ppGpp maintains its influence despite decreasing in absolute amounts.

## The impact of ppGpp on photosynthesis under nitrogen deprivation

Our work also reveals the wide-ranging physiological effects of ppGpp on photosynthetic activity during nitrogen deprivation. These effects are reminiscent of the photosynthetic changes that occur during senescence (*Krieger-Liszkay et al., 2019*), and are remarkably similar to those found using the ectopic overaccumulation of ppGpp in *Arabidopsis*, moss and algae (*Maekawa et al., 2015*; *Sugliani et al., 2016*; *Imamura et al., 2018*; *Honoki et al., 2018*; *Avilan et al., 2021*; *Harchouni et al., 2021*). However, compared to these studies our more detailed analysis here highlights first the global effect of ppGpp on chloroplast transcript abundance (*Figure 5A*), and second the strong effect of ppGpp on certain photosynthetic proteins such as PsbO from PSII, PetA from Cyt $b_6f$, and RbcL (*Figure 3A, B*). ppGpp controls the abundance of these proteins via the downregulation of chloroplast transcript abundance (*Figure 5A*), reducing the quantity of transcripts available for translation as well as translation capacity via a likely reduction in rRNA transcription. The downregulation of transcript abundance may involve a GTP concentration-dependent inhibition of transcription as discussed above. However, the strong reduction in subunits from complexes that are usually stable like Cyt $b_6f$ suggests that more active processes could also be involved. This idea is supported by previous work showing the high stability of Cyt $b_6f$ in tobacco (*Hojka et al., 2014*), the selective loss of Cyt $b_6f$ during natural senescence (*Krieger-Liszkay et al., 2019*; *Roberts et al., 1987*), and the protease-mediated degradation mechanism for Cyt $b_6f$ and Rubisco that occurs under nitrogen limitation in *Chlamydomonas* (*Majeran et al., 2000*; *Wei et al., 2014*). Another argument supporting the idea that ppGpp may act on protein stability is the contrast between reduced transcript and stable protein levels for the nuclear-encoded photosynthetic genes Lhcb1 and Lhca1 in low ppGpp plants (*Figures 3A, B and 5C*, *Figure 5—source data 3*). This contrast suggests that ppGpp could promote the degradation of these proteins during nitrogen limitation.

PSII architecture was also modified in a ppGpp-dependent manner during nitrogen limitation with an increase in the ratio of PSII antenna to RC (*Figure 3A–D*), causing an increase in basal fluorescence and contributing to the decrease in PSII maximal efficiency (Fv/Fm) (*Figure 2A, D*). While this phenomenon has similarities to photoinhibition, our data suggest that ppGpp does not increase PSII RC photoinactivation, but instead inhibits the resynthesis of PSII RC via the inhibition of chloroplast transcription. Due to the rapid degradation of photoinactivated PSII RC this leads to an increase in the ratio of PSII antenna to RC.

## ppGpp is implicated in photoprotection during nitrogen limitation

During nitrogen deficiency plants must remobilize nitrogen to drive acclimation processes. At the same time, growth arrest leads to a strong decrease in demand for the products of photosynthesis which has the potential to lead to overexcitation and ROS production. Therefore, in addition to remobilizing nitrogen, the plant must also downsize the photosynthetic machinery in a controlled fashion to avoid producing damaging levels of ROS. Here, we show that ppGpp promotes a reduction in the photosynthetic machinery, a major nitrogen store, and at the same time plays a role in protecting against excess ROS production. We show that ROS increases in the wild type in response to nitrogen deprivation, and that defects in ppGpp biosynthesis lead to greater increases in ROS and cell death (*Figure 1B, D*). The limited ROS increase in the wild type is in agreement with a recent study also showing ROS accumulation in the leaves of nitrogen starved plants (*Safi et al., 2021*). Chloroplasts are the major source of ROS in photosynthetic plant leaves (*Rogers and Munné-Bosch, 2016*; *Domínguez and Cejudo, 2021*). The role for ppGpp signaling in the prevention of ROS overaccumulation and tissue death during nitrogen limitation (*Figure 1A–D*) is therefore very likely related to its role in downregulating photosynthetic activity (*Figure 2*, *Figure 2—figure supplement 2*, *Figure 2—figure supplement 4*). As part of this process, ppGpp signaling alters the stoichiometry of PSII components, reducing excitation pressure on PSII and thus electron flow in the photosynthetic chain. Note that the reduced Rubisco levels we observe under nitrogen limitation (*Figure 3A, B*) suggest that PSI may be acceptor side limited due to reduced activity of the Calvin–Benson cycle, which could lead to increased

superoxide formation by reduction of molecular oxygen at PSI. Indeed, others have proposed that the reduction in levels of active PSII RC during photoinhibition can protect PSI against overexcitation damage (*Tikkanen et al., 2014*). Interestingly, the ppGpp-mediated alteration of PSII architecture appears to be independent of the global reduction in photosynthetic proteins during nitrogen limitation: photosynthetic proteins are downregulated in low ppGpp plants without major alterations in PSII activity (*Figures 3A, B and 2C, D*). On the basis of these results, we suggest that one of the functions of ppGpp signaling may therefore be to optimize nitrogen remobilization, notably via the degradation of the most abundant leaf protein Rubisco, modulating at the same time the photosynthetic machinery to prevent overexcitation and photodamage. The most straightforward explanation based on the evidence presented here is that ppGpp acts through a general and coordinated downregulation of chloroplast gene expression, and further evidence would be needed to determine whether ppGpp is also able to play a more specific role in remodeling photosynthesis.

We demonstrate a new physiological role for ppGpp signaling and offer new insights into the molecular mechanisms underlying its action. The wide range of stresses known to induce ppGpp accumulation (*Takahashi et al., 2004*) and the association between stress and overexcitation of the photosynthetic electron transport chain (*Bechtold and Field, 2018*), together suggest that ppGpp signaling may allow plants to anticipate and acclimate to harsh environmental conditions that would otherwise lead to overexcitation of the photosynthetic machinery and oxidative stress. At the same time, our work also highlights important questions about ppGpp function that are not yet fully answered: how does nitrogen limitation activate ppGpp signaling? Does ppGpp target a single process (transcription) or several? and Does ppGpp signaling interact with known photoprotective pathways (*Pinnola and Bassi, 2018*; *Malnoë, 2018*)?

# Materials and methods

**Key resources table**

| Reagent type (species) or resource | Designation | Source or reference | Identifiers | Additional information |
|---|---|---|---|---|
| Genetic reagent (*Arabidopsis thaliana*) | Col-0 | Nottingham *Arabidopsis* Stock Centre (NASC) | WT1, Columbia, N1093 (NASC) | |
| Genetic reagent (*Arabidopsis thaliana*) | qrt1-2 | Nottingham *Arabidopsis* Stock Centre (NASC), *Copenhaver et al., 2000* | WT2, N8846 (NASC) | Col-3 ecotype |
| Genetic reagent (*Arabidopsis thaliana*) | qrt1-2/rsh1-1 | Nottingham *Arabidopsis* Stock Centre (NASC), *Sugliani et al., 2016*; *Sessions et al., 2002* | TDNA insertion SAIL_391_E11, N818025 (NASC) | *RSH* mutant |
| Genetic reagent (*Arabidopsis thaliana*) | qrt1-2/rsh2-1 | Nottingham *Arabidopsis* Stock Centre (NASC), *Sugliani et al., 2016*; *Sessions et al., 2002* | TDNA insertion SAIL_305_B12, N814119 (NASC) | *RSH* mutant |
| Genetic reagent (*Arabidopsis thaliana*) | qrt1-2/rsh3-1 | Nottingham *Arabidopsis* Stock Centre (NASC), *Sugliani et al., 2016*; *Sessions et al., 2002* | TDNA insertion SAIL_99_G05, N862398 (NASC) | *RSH* mutant |
| Genetic reagent (*Arabidopsis thaliana*) | qrt1-2/rsh2-1 rsh3-1 | *Sugliani et al., 2016* | DM-23 | *RSH* mutant |
| Genetic reagent (*Arabidopsis thaliana*) | qrt1-2/rsh$_{QM}$ | *Sugliani et al., 2016* | rsh1-1, rsh2-1, rsh3-1, crsh-ami/Qmaii | *RSH* mutant |
| Genetic reagent (*Arabidopsis thaliana*) | Col-0/OX:RSH1 | *Sugliani et al., 2016* | OX:RSH1-GFP (10.4) | Overexpression line |
| Genetic reagent (*Arabidopsis thaliana*) | qrt1-2/rsh2-1 rsh3-1 pRSH3:RSH3 | *Sugliani et al., 2016* | C11 | Complementation line |
| Genetic reagent (*Arabidopsis thaliana*) | qrt1-2/rsh2-1 rsh3-1 pRSH3:RSH3 | *Sugliani et al., 2016* | C43 | Complementation line |

*Continued on next page*

*Continued*

| Reagent type (species) or resource | Designation | Source or reference | Identifiers | Additional information |
|---|---|---|---|---|
| Genetic reagent (*Arabidopsis thaliana*) | *qrt1-2/rsh2-1 rsh3-1* pRSH3:RSH3 | *Sugliani et al., 2016* | C131 | Complementation line |
| Genetic reagent (*Arabidopsis thaliana*) | *qrt1-2/rsh2-1 rsh3-1* pRSH3:RSH3 | *Sugliani et al., 2016* | CX3 | Complementation line |
| Genetic reagent (*Arabidopsis thaliana*) | Col-0/pOP:MESH | *Sugliani et al., 2016* | MESH | DEX inducible MESH |
| Genetic reagent (*Arabidopsis thaliana*) | *qrt1-2/rsh1-1* pRSH1:RSH1 | This study | C110 | Materials and methods: creation of *rsh1-1* complementation lines |
| Antibody | anti-COXII (rabbit polyclonal) | Agrisera | AS04 053A | Dilution (1:2000) |
| Antibody | anti-LHCA1 (rabbit polyclonal) | Agrisera | Ref. AS01 005 | Dilution (1:2000) |
| Antibody | anti-LHCB1 (rabbit polyclonal) | Agrisera | Ref. AS01 004 | Dilution (1:2000) |
| Antibody | anti-PBA1 (rabbit polyclonal) | Abcam | Ref. ab98861 | Dilution (1:2000) |
| Antibody | anti-PetA (rabbit polyclonal) | Agrisera | Ref. AS08 306 | Dilution (1:2000) |
| Antibody | anti-PsaD (rabbit polyclonal) | Agrisera | Ref. AS04 046 | Dilution (1:2000) |
| Antibody | anti-PsbA (rabbit polyclonal) | Agrisera | Ref. AS05 084 | Dilution (1:2000) |
| Antibody | anti-PsbO (rabbit polyclonal) | Agrisera | Ref. AS05 092 | Dilution (1:2000) |
| Antibody | anti-PsbS (rabbit polyclonal) | Agrisera | Ref. AS09 533 | Dilution (1:1000) |
| Antibody | anti-PTOX (rabbit polyclonal) | Uniplastomic | Kindly provided by X.Johnson | Dilution (1:2000) |
| Antibody | anti-SAG12 (rabbit polyclonal) | Agrisera | Ref. AS14 2771 | Dilution (1:2000) |
| Chemical compound, drug | 13C-ppGpp | Kindly provided by J.Bartoli and E.Bouveret | | Internal standard for ppGpp quantification |
| Chemical compound, drug | 13C-GTP | Sigma-Aldrich | 710687 | Internal standard for GTP quantification |
| Chemical compound, drug | 15-HEDE | Cayman Chemical | Item No. 37700 | Internal standard for HOTE quantification |
| Commercial assay or kit | Oasis WAX 1 cc Vac Cartridge | Waters | Ref. 186002491 | Nucleotide quantification |
| Chemical compound, drug | Nucleozol | Macherey Nagel | Ref. 740404.200 | RNA extraction |
| Chemical compound, drug | 4-Bromoanisole | Sigma-Aldrich | B56501 | RNA extraction |
| Commercial assay or kit | Clean & Concentrator-25 kit | Zymo Research | Cat. No. R1017 | RNA extraction |
| Commercial assay or kit | Ribo-Zero rRNA Removal Kit (Plant) | Illumina | Ref. MRZPL116 | RNA treatment |
| Other | Open FluorCam | Photon System Instruments | FC 800-O/2020-GFP | Chlorophyll fluorescence |
| Software, algorithm | R | *R Development Core Team, 2020* | | Data analysis |
| Software, algorithm | ggplot2 package | *Wickham, 2009* | | Data analysis |
| Software, algorithm | Rmisc package | *Hope, 2013* | | Data analysis |
| Software, algorithm | boot package | *Canty and Ripley, 2021*; *Davison and Hinkley, 1997* | | Data analysis |
| Software, algorithm | rstatix package | *Kassambara, 2021* | | Data analysis |

*Continued on next page*

*Continued*

| Reagent type (species) or resource | Designation | Source or reference | Identifiers | Additional information |
|---|---|---|---|---|
| Software, algorithm | Rcompanion package | *Salvatore, 2021* | | Data analysis |
| Software, algorithm | prepare_gene_ontology.pl script | This study | | Data analysis, available at: https://github.com/cecile-lecampion/gene-ontology-analysis-and-graph |

## Plant materials and growth conditions

*Arabidopsis thaliana* mutant lines, overexpressors and complementation lines and respective wild-type controls are listed in the Key Resources table. In each experiment, the seeds for each line were derived from the same batch of plants grown together. For growth on plates, seeds were surface sterilized with 75% ethanol, rinsed with 100% ethanol, dried and placed in a 10 by 10 grid pattern on square culture plates containing 45 ml of nitrogen replete (+N) (0.5× Murashige and Skoog salts [Caisson Labs], 1% sucrose, 0.5 g l$^{-1}$ MES, and 0.4% Phytagel [Merck Sigma-Aldrich], adjusted to pH 5.7 with KOH) or nitrogen limiting (−N) (+N medium diluted 1/25 in 0.5× Murashige and Skoog medium without nitrogen [Caisson Labs], 0.5 g l$^{-1}$ MES and 0.4% Phytagel [Merck Sigma-Aldrich], adjusted to pH 5.7 with KOH) growth medium (for detailed medium composition see *Supplementary file 1*). Plates were placed at 4°C for 2 days in the darkness, and then transferred to a 16-hr light (at 22°C)/8-hr darkness (at 19.5°C) photoperiod with 80 µmol photons m$^{-2}$ s$^{-1}$ lighting. For growth in quartz sand, seeds were germinated in soil, and then transferred into pots containing quartz sand at 7 days after germination. The plants were then grown under a 8-hr light (at 22°C)/16-hr dark (at 19.5°C) photoperiod at 18/22°C with 110 µmol photons m$^{-2}$ s$^{-1}$ lighting. Plants were treated weekly with a complete nutrient solution. After 50 days the pots were rinsed, and then treated weekly with 0.5× Murashige and Skoog medium with or without nitrogen for 3 weeks.

## Creation of *rsh1-1* complementation lines

The 9.1-kb genomic *RSH1* sequence including the 3′ untranslated region, 5′ untranslated region, and 3 kb of upstream sequence containing the promoter was amplified from *Arabidopsis* genomic DNA using Phusion polymerase (New England Biolabs) using flanking primers and then the primers RSH1-F (5′-TCCGTCTTGTCTGAATCAGCT-3′) and RSH1-R (5′-TTTCTAGATTTACTTTGGTTTTGTCCA-3′) with attB1/attB2 adapters. The resulting PCR product was then introduced by Invitrogen BP Gateway recombination (Thermo Fisher) into pDONR207. The entry clone was confirmed by sequencing and recombined by Invitrogen LR Gateway recombination into the binary vector pGWB410 (*Nakagawa et al., 2007*) which carries a kanamycin resistance cassette for selection in plants. The resulting construct was transferred into *Agrobacterium tumefaciens* (strain GV3101) and used to transform *rsh1-1* plants by floral dipping.

## Autoluminescence imaging and quantification of HOTEs

Peroxidated lipids were visualized by autoluminescence imaging as described previously (*Birtic et al., 2011*) and similar results were observed in four biological replicates. The intensity of the leaf autoluminescence signal is proportional to the amount of lipid peroxides present in the sample, the slow spontaneous decomposition of which produces luminescent species. For quantification of HOTEs, lipids were extracted from about 300 mg of seedlings in methanol/chloroform and analyzed by HPLC–UV as detailed elsewhere (*Montillet et al., 2004*; *Shumbe et al., 2017*).

## Nucleotide triphosphate quantification

Nucleotides were extracted from about 150 mg of seedlings, and quantified by HPLC–MS/MS using stable isotope labeled ppGpp and GTP standards as described previously (*Bartoli et al., 2020*).

## Chlorophyll fluorescence measurements

Plants were dark adapted for 20 min and chlorophyll fluorescence was measured in a Fluorcam FC 800-O imaging fluorometer (Photon System Instruments). PSII maximum quantum yield (Fv/Fm) was calculated as (Fm − Fo)/Fm. For relative electron transfer rate (ETR) measurements plants were

exposed to 2-min periods of increasing actinic light intensity, with PSII quantum yield measurements at the end of each period. Relative ETR was calculated as quantum yield × light intensity (photosynthetic photon flux density). Photosynthetic photon flux density was measured using a Universal Light Meter 500 (ULM-500, Walz). All experiments on photosynthetic parameters were repeated two to five times with similar results. Steady-state 77 K chlorophyll fluorescence measurements were obtained from frozen seedling powder suspended in 85% (wt/vol) glycerol, 10 mM HEPES (4-(2-hydroxyethyl)-1 -piperazineethanesulfonic acid), pH 7.5 as described previously (*Galka et al., 2012*).

## Protein extraction and immunoblotting

Protein extraction and immunoblotting were performed on 100 bulked seedlings as described previously (*Sugliani et al., 2016*), with the addition of a protein precipitation step with 20% trichloroacetic acid after protein extraction. The precipitation step was necessary to remove compounds in nitrogen-limited seedlings that interfered with the bicinchoninic acid assay for determining protein concentration. The following primary antibodies were used against CoxII (Agrisera; polyclonal; ref AS04 053A), Lhca1 (Agrisera; polyclonal, ref AS01 005), Lhcb1 (Agrisera; polyclonal, ref AS01 004), Pba1 (Abcam; polyclonal; ref ab98861), PetA (Agrisera; polyclonal, ref AS08 306), PsaD (Agrisera; polyclonal; ref AS04 046), PsbA (Agrisera; polyclonal, ref AS05 084), PsbO (Agrisera; polyclonal, ref AS05 092), PsbS (Agrisera; polyclonal, ref AS09 533), PTOX (Uniplastomic, Biviers, France; kindly provided by Xenie Johnson), and SAG12 (Agrisera; polyclonal; ref AS14 2771).

For the analysis of thylakoidal proteins, 100 bulked seedlings were harvested and then ground in liquid nitrogen. Proteins were extracted in an extraction buffer adapted from *Pesaresi, 2011* (0.4 M sucrose, 10 mM NaCl, 5 mM MgCl$_2$, 10 mM tricine KOH, pH 7.5, 100 mM ascorbate, 0.2 mM phenylmethylsulfonyl fluoride, 5 mM aminocaproic acid). Samples were centrifuged at 1000 × *g* for 5 min at 4°C, then the pellets were resuspended in extraction buffer. The samples were centrifuged again at 1000 × *g* for 5 min at 4°C, and the resulting chloroplast-enriched pellets suspended in lysis buffer as previously described (*Chen et al., 2016*) (10 mM Tricine–NaOH, pH 7.8) and incubated for 30 min. The samples were centrifuged at 5000 × *g* for 5 min at 4°C, and the pellets containing the membrane fraction were resuspended in lysis buffer. Samples were centrifuged at 1000 × g for 5 min at 4°C, then the pellets were resuspended in extraction buffer (100 mM Tris, pH 6.8, 20% glycerol, 10% sodium dodecyl sulfate). The samples were heated to 40°C for 5 min and then centrifuged at 1300 × g for 10 min at room temperature. The supernatant was recovered, and the samples were normalized on the amount of chlorophyll.

## Chlorophyll quantification

Pigments were extracted in 80% acetone then separated and quantified by HPLC–UV as described previously (*Campoli et al., 2009*).

## RNA sequencing

RNA sequencing was performed with three biological replicates on 100 bulked seedlings per line grown on +N medium for 8 days or −N medium for 12 days. RNA was extracted from frozen seedling powder with Nucleozol (Macherey-Nagel) with 4-bromoanisole to reduce DNA and anthocyanin contamination. Total RNA was cleaned and concentrated using RNA Clean & Concentrator-25 (Zymo Research) according to the manufacturer's instructions. Genomic DNA was removed by treatment with DNase. RNA-seq libraries were constructed by the POPS platform (IPS2) using the TruSeq Stranded mRNA library prep kit (Illumina) with RiboZero plant (Illumina). Libraries were sequenced in single end (SE) with a read length of 75 bases for each read on a NextSeq500 (Illumina). Approximately 30 million reads by sample were generated. Adapter sequences and bases with a *Q*-score below 20 were trimmed out from reads using Trimmomatic (v0.36) (*Bolger et al., 2014*) and reads shorter than 30 bases after trimming were discarded. Reads corresponding to rRNA sequences were removed using sortMeRNA (v2.1) (*Kopylova et al., 2012*) against the silva-bac-16s-id90, silva-bac-23s-id98, silva-euk-18s-id95, and silva-euk-28s-id98 databases. Read quality checks were performed using FastQC (Version 0.11.5) (*Andrews, 2010*). The raw data (fastq) were then aligned against the *Arabidopsis* transcriptome ( Araport11_cdna_20160703_representative_gene_model.fa) concatenated with noncoding RNA ( TAIR10.ncrna.fa) using Bowtie2 (version 2.2.9) (*Langmead and Salzberg, 2012*). Default parameters were used. Reads were counted using a modified version of a command line previously described

(*Van Verk et al., 2013*). Differential expression analysis was performed with SARTools (version 1.7.3) (*Varet et al., 2016*) using edgeR. Study details and Fastq files were deposited at the European Nucleotide Archive (https://www.ebi.ac.uk/ena) under accession number PRJEB46181. Workflow and analysis reports for RNA-seq data analysis are provided in *Supplementary file 2*.

## Data analysis

The majority of analysis was conducted in R (*R Development Core Team, 2020*) and custom annotated R markdown scripts are provided in *Supplementary file 3*. Graphs were produced using the package ggplot2 (*Wickham, 2009*) with confidence intervals determined for normally distributed data using the Rmisc package (*Hope, 2013*), and nonparametric data with boostrap confidence interval calculation using the boot package (*Canty and Ripley, 2021*; *Davison and Hinkley, 1997*). Individual plants were counted as biological replicates, and experiments were repeated at least two times with similar results. For the timecourse in *Figure 3A* and the nitrogen deprivation performed on mature plants in *Figure 2—figure supplement 2E* single experimental replicates were performed. Statistical analyses were performed on normally distributed data using analysis of variance with post hoc Tukey test and nonparametric data using the Kruskal–Wallis test with post hoc Dunn test in the rstatix package (*Kassambara, 2021*). For categorical data (*Figure 1B, C*) significance was calculated using a proportion test and post hoc Fisher's test using the Rstatix and Rcompanion packages (*Kassambara, 2021*; *Salvatore, 2021*). All statistical tests included adjustments for multiple comparisons.

GO enrichment analysis for the RNAseq data was performed using a custom prepare_gene_ontology.pl script (https://github.com/cecile-lecampion/gene-ontology-analysis-and-graph; *Romand, 2021* copy archived at swh:1:rev:dd4f832a6d1d51c5b72a3b52dff9c1c7e964fe95, commit c170f8e90323f3061d44a87fcb76d6b0e8b00f63) which automatically uses PANTHER and REVIGO for the identification and simplification of enriched GO terms according to the procedure proposed by *Bonnot et al., 2019*. Results were plotted using ggplot2.

## Acknowledgements

We thank colleagues at the LGBP, Xenie Johnson and Wojciech Nawrocki for critical discussion of the manuscript. We thank Julia Bartoli and Emmanuelle Bouveret for supplying isotope labeled ppGpp. Nucleotide measurements were performed on the IJPB Plant Observatory technological platform, and transcriptomics on the POPS platform which are both supported by Saclay Plant Sciences-SPS (ANR-17-EUR-0007). The work was funded by Agence Nationale de la Recherche (ANR-17-CE13-0005).

## Additional information

### Funding

| Funder | Grant reference number | Author |
| --- | --- | --- |
| Agence Nationale de la Recherche | ANR-17-CE13-0005 | Benjamin Field |
| Agence Nationale de la Recherche | ANR-17-EUR-0007 | Sylvie Citerne Jose Caius |

The funders had no role in study design, data collection, and interpretation, or the decision to submit the work for publication.

### Author contributions

Shanna Romand, Conceptualization, Formal analysis, Investigation, Methodology, Visualization, Writing – original draft, Writing – review and editing; Hela Abdelkefi, Investigation, Methodology, Writing – review and editing; Cécile Lecampion, Data curation, Resources, Software, Visualization, Writing – review and editing; Mohamed Belaroussi, Melanie Dussenne, Jose Caius, Investigation, Methodology; Brigitte Ksas, Sylvie Citerne, Formal analysis, Investigation, Methodology; Stefano D'Alessandro, Conceptualization, Investigation, Methodology, Writing – review and editing; Hatem Fakhfakh, Supervision, Writing – review and editing; Stefano Caffarri, Conceptualization, Methodology;

Michel Havaux, Conceptualization, Investigation, Methodology, Supervision, Writing – review and editing; Ben Field, Conceptualization, Data curation, Formal analysis, Funding acquisition, Investigation, Methodology, Project administration, Supervision, Validation, Visualization, Writing – original draft, Writing – review and editing

**Author ORCIDs**
Cécile Lecampion  http://orcid.org/0000-0002-7862-517X
Ben Field  http://orcid.org/0000-0003-2142-4606

**Decision letter and Author response**
Decision letter https://doi.org/10.7554/eLife.75041.sa1
Author response https://doi.org/10.7554/eLife.75041.sa2

## Additional files

### Supplementary files
- Supplementary file 1. Media composition.
- Supplementary file 2. RNA-seq analysis reports. Related to *Figure 4*.
- Supplementary file 3. R markdown scripts. Related to all figures.
- Transparent reporting form

### Data availability
All data presented in this study are included in the manuscript and supporting files Source Data files have been provided for Figures 1-5 (+ supplements). Sequencing data have been deposited at the European Nucleotide Archive under accession number PRJEB46181.

The following dataset was generated:

| Author(s) | Year | Dataset title | Dataset URL | Database and Identifier |
|---|---|---|---|---|
| Romand S, Lecampion C, Field B | 2021 | Effects of ppGpp on gene expression during nitrogen deprivation in Arabidopsis | https://www.ebi.ac.uk/ena/browser/view/PRJEB46181?show=reads | European Nucleotide Archive, PRJEB46181 |

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
