## [Editor Report]

The authors provide genetic evidence showing that the unusual nucleotide, guanosine tetraphosphate (ppGpp), modulates plastidial gene expression and photosynthetic electron transfer upon nitrogen starvation in Arabidopsis. The results provide further insight into functional importance of the plastidial stringent response in plant physiology.

---

## [Decision Letter]

**Decision letter after peer review:**

Thank you for submitting your article "A ppGpp-mediated brake on photosynthesis is required for acclimation to nitrogen limitation in Arabidopsis" for consideration by *eLife*. Your article has been reviewed by 3 peer reviewers, and the evaluation has been overseen by a Reviewing Editor and Jürgen Kleine-Vehn as the Senior Editor. The following individuals involved in review of your submission have agreed to reveal their identity: Sujith Puthiyaveetil (Reviewer #3); Alizée Malnoë (Reviewer #4).

Essential revisions:

*Reviewer #1 (Recommendations for the authors):*

This manuscript entitled "A ppGpp-mediated brake on photosynthesis is required for acclimation to nitrogen limitation in Arabidopsis " by Romand et al. described ppGpp accumulation is necessary for acclimation to nitrogen starvation in a model plant Arabidopsis. The authors revealed that ppGpp accumulation leads to remodeling the photosynthetic electron transport chain to downregulate photosynthetic activity. It seems to be reasonable that the ppGpp signal works for protecting plant cells from oxidative stress. Further, the authors also clarified that ppGpp accumulation affects patterns of chloroplast gene expression during nitrogen starvation. The crucial findings based on their hard work should be highly appreciated.

On the other hand, the authors left not a few points to be amended in Materials and methods section. Thereby the manuscript cannot be accepted at present.

*Reviewer #2 (Recommendations for the authors):*

1. Lines 189-194. The attempt to distinguish between in vivo and in vitro nature of the measurements is not clear. Please note Fv/Fm is an in vivo measurement whether it is done on plants grown on plates or quarts sand.

2. Line 211. "energetic pressure" => "excitation pressure".

3. The protein levels in Figure 3 may reflect a decrease in de novo protein synthesis as wells as an increased degradation. Please qualify the discussion as such.

4. Lines 375-388. References to Figure 5- supplement 1 and 2 do not correspond with the actual names of Figure 5 supplements. Please rename/rephrase the supplement figure legends or main text.

5. Line 498. What is the data in support of the increase in antenna to RC transcript ratio? In wild type both RC and antenna transcript levels decrease under nitrogen deficiency.

6. Please describe the genotype of the quadruple mutant in methods.

*Reviewer #4 (Recommendations for the authors):*

– The reason(s) for comparing 8-day-old, non-stressed plants to 12-day-old, stressed plants throughout most of the article is not provided in the text. Please explain this choice.

– Page 8, Figure 2, panels A and D (also for figure supplement 1 page 39): the Fv/Fm parameter, while informative, can cover several different mechanisms (see Malnoë, Environmental and Experimental Botany, 2018). Please provide Fo and Fm values at least in the source files for these figures. Are the data presented measured on whole seedlings (i.e. including cotyledons)? It would be preferable to separate the data on true leaves from the cotyledons as the metabolism from these two tissue types is vastly different.

– Line 293, energetic uncoupling could be an indirect effect due to PsbA degradation under -N when ppGpp present. The lower Fv/Fm being due to a high Fo from disconnected antenna, rather than due to damaged centers displaying a high Fo (and low Fm due to qI). Revise statement page 21, lane 499 accordingly. Is the data in Fig3D normalized? Line 508-513, another possibility is that altered ppGpp/GTP ratio affects turnover of D1 (see Kato et al. Plant Physiology, 2018 Vol. 178, pp. 596-611). We would therefore suggest to temper down the statement line 525 that "ppGpp signalling triggers the controlled photoinhibition of PSII" and line 528 "functions as a photoprotective mechanism". An alternative explanation could be that ppGpp signaling is to enable scavenging of N, and that accumulation of photosynthetic complexes in -N causing imbalance and ROS is an indirect consequence of low ppGpp. Also provide discussion in that section about statement line 98 that there is a higher new leaf initiation rate in low ppGpp lines.

– Page 12, line 300: what about RNA-seq data from the rsh1-1 mutant? It seems an important control that would confirm claim that ppGpp downregulates chloroplast gene expression (page 15, line 348).

­ – Line 320, it is stated that with less ppGpp, nuclear encoded genes for photosynthesis/chloroplast activity are downregulated, yet protein amount are the same, could you discuss turnover rate being possibly slower? Page 15, line 354, "ppGpp acts specifically within the chloroplast" but then how do you explain the lower nuclear encoded genes expression when there is less ppGpp?

– The overexpressor line "OX:RSH1" is under the 35S promoter which has been shown to possibly yield to silencing (see Grefen et al. Plant Journal, 2010 Vol. 64, pp. 355-365). Have you confirmed the higher expression of RSH1 in your various experiments? A lower expression may explain the higher ppGpp levels than expected shown in Figure 1F and discussed line 453.

– Page 18, line 403: explain "relaxed", these bacterial mutants were lacking ppGpp, correct and this work showed that a role for ppGpp inhibiting RNA synthesis? Please provide more explanation.

[Editors' note: further revisions were suggested prior to acceptance, as described below.]

Thank you for resubmitting your work entitled "A ppGpp-mediated brake on photosynthesis is required for acclimation to nitrogen limitation in Arabidopsis" for further consideration by *eLife*. Your revised article has been evaluated by Jürgen Kleine-Vehn (Senior Editor) and a Reviewing Editor.

The manuscript has been reviewed by original reviewers. They judge that the manuscript has been improved but there are some remaining issues that need to be addressed before publication. The comments are reliable, so that we further request you to revise the manuscript by taking into account the editor's and the reviewers' comments, which are outlined below:*Editor:*

The title of the journal should provide a clear indication of the biological system and it should avoid specialist abbreviations and acronyms where possible. Thus, the editor judgeds that use of "ppGpp" needs to be avoided in the title, and ask the authors to change the title such as "A bacterial signaling molecule-mediated brake~".

*Reviewer 1:*

The authors have provided a detailed response to my comments and have addressed most of my concerns by revising the manuscript. The authors now note, as a reviewer suggested, that the low Fv/Fm values of the ppGpp over accumulator is due to a high Fo values. Given this, the authors have considerably toned down their argument about ppGpp being an active remodeler of PSII functional assembly, but they still allude to this possibility elsewhere (lines 562-570). I don't protest this hugely but there seems to be a unanimous feeling among the three reviewers that the authors also mention (after line 570) an alternate hypothesis that the decreased PSII activity could simply be a consequence of a coordinated decrease in plastid protein synthesis by ppGpp rather than active regulation of PSII function by ppGpp. A similar downregulation of activity may be apparent for PSI but of course they haven't measured it.

---

## [Author Response]

Essential revisions:Reviewer #2 (Recommendations for the authors):1. Lines 189-194. The attempt to distinguish between in vivo and in vitro nature of the measurements is not clear. Please note Fv/Fm is an in vivo measurement whether it is done on plants grown on plates or quarts sand.

Yes, this was lab jargon and was not clear. We have corrected it.

2. Line 211. "energetic pressure" => "excitation pressure".

Corrected.

3. The protein levels in Figure 3 may reflect a decrease in de novo protein synthesis as wells as an increased degradation. Please qualify the discussion as such.

We agree fully, and we have modified the phrase where we say this to make it more clear from:

“ppGpp may control the abundance of these proteins via the downregulation of chloroplast transcript abundance (Figure 5A). “

To:

“ppGpp may influence the abundance of these proteins via the downregulation of chloroplast transcript abundance (Figure 5A), reducing the quantity of transcripts available for translation as well as translation capacity via a likely reduction in rRNA transcription. The downregulation of transcript abundance may involve a GTP concentration dependent inhibition of transcription as discussed above.”

We have also toned down the subsequent discussion about the potential role of degradation mechanisms as discussed above.

4. Lines 375-388. References to Figure 5- supplement 1 and 2 do not correspond with the actual names of Figure 5 supplements. Please rename/rephrase the supplement figure legends or main text.

This has been corrected. Please note that for technical reasons *ELife* could not include these multipage figures as figure supplements. The extended figures are now found in Figure 5- source data 3.

5. Line 498. What is the data in support of the increase in antenna to RC transcript ratio? In wild type both RC and antenna transcript levels decrease under nitrogen deficiency.

Thank you for spotting this- in line 498 we made an error. Figure 5C does not support this statement. The correct figure would be Figure 5—figure supplement 1, and this actually shows the inverse- the nucleus (antenna) transcript levels decrease more on average than the chloroplast transcripts during nitrogen deficiency for PSII and the other complexes. We have therefore removed the reference to transcript ratios.

6. Please describe the genotype of the quadruple mutant in methods.

We used 14 different plant lines in this study and their genotypes were described in a table in supplementary file 1, and now this information is in the *eLife* key resources table.

Reviewer #3 (Recommendations for the authors):– The reason(s) for comparing 8-day-old, non-stressed plants to 12-day-old, stressed plants throughout most of the article is not provided in the text. Please explain this choice.

We chose 12 day old stressed plants because at this time-point we saw the clearest differences between the mutants and the wild-type controls. Eight day-old non stressed plants were chosen as a control because they are at a similar developmental stage (size, leaf number). The +N control is used principally to show that under normal growth conditions there is little or no difference between the lines for the phenotypes tested. We have added an explanation in the legend of Figure 1.

– Page 8, Figure 2, panels A and D (also for figure supplement 1 page 39): the Fv/Fm parameter, while informative, can cover several different mechanisms (see Malnoë, Environmental and Experimental Botany, 2018). Please provide Fo and Fm values at least in the source files for these figures. Are the data presented measured on whole seedlings (i.e. including cotyledons)? It would be preferable to separate the data on true leaves from the cotyledons as the metabolism from these two tissue types is vastly different.

We agree that these are important values, and we confirm that the drop we observe in Fv/Fm are due to an increase in Fo relative to Fm (or Fv) during nitrogen deficiency. This is consistent with our other data showing that there are disconnected antennas, and we previously noted it as an effect of ppGpp overaccumulation in Sugliani et al. 2016. We now provide the Fo and Fm values in the source data, and indicate that in the text that an Fo rise is behind the drop in Fv/Fm.

Data are presented on a whole seedling basis (as shown in Figure 2B). You will note that there are indeed differences in the responses of true leaves and cotyledons. At the same time our experiments on mature plants show that the ppGpp-mediated Fv/Fm response to nitrogen deficiency is not specific to seedlings (Figure 2—figure supplement 1). We think that separating out and resolving these differences would be interesting and could reveal tissue specific roles for ppGpp signalling, however we think that this is beyond the scope of the present work.

– Line 293, energetic uncoupling could be an indirect effect due to PsbA degradation under -N when ppGpp present. The lower Fv/Fm being due to a high Fo from disconnected antenna, rather than due to damaged centers displaying a high Fo (and low Fm due to qI). Revise statement page 21, lane 499 accordingly.

We agree with the reviewer, and in fact we think that this was our meaning. We have re-written this part of the discussion in line with this comment and the comments of Reviewer 3. We highlight the similarities and differences between photoinhibition and the nitrogen limitation induced ppGpp dependent drop in Fv/Fm.

Is the data in Fig3D normalized?

Yes. We now mention this in the legends of 3D and Figure 2- supplement 2.

Line 508-513, another possibility is that altered ppGpp/GTP ratio affects turnover of D1 (see Kato et al. Plant Physiology, 2018 Vol. 178, pp. 596-611). We would therefore suggest to temper down the statement line 525 that "ppGpp signalling triggers the controlled photoinhibition of PSII" and line 528 "functions as a photoprotective mechanism". An alternative explanation could be that ppGpp signaling is to enable scavenging of N, and that accumulation of photosynthetic complexes in -N causing imbalance and ROS is an indirect consequence of low ppGpp.

We have modified our discussion of the similarities with photoinhibition which was not clear.

Furthermore, while we agree that ppGpp signalling is likely to have a role in remobilising N during nitrogen deficiency we do not think that this excludes or can be split from a role in protecting against overexcitation of the photosynthetic machinery, the production of ROS and cell death. We have modified the discussion to make this point clearer. Below we provide a rationale for our position.

During nitrogen deficiency the plant requires nitrogen, but also experiences a strong decrease in demand for the products of photosynthesis (energy and sugar) because growth is arrested. Therefore, in addition to remobilising nitrogen the plant must also downsize the photosynthetic machinery to avoid overexcitation. The two needs can be met at the same time because downsizing the photosynthetic machinery will remobilise nitrogen. The photosynthetic chain must be downsized in a controlled manner and with coordination between the nuclear and chloroplast genomes to avoid ROS overaccumulation. This could occur by simply reducing but preserving the same functioning of the whole chain. However, our results show that this is not the case: PSII architecture changes (as measured by 77K (Figure 3D), chlorophyll (Figure 3C), immunoblot (Figure 3A/B) and also Fv/Fm which drops from the early stages of nitrogen deprivation (day 8) to the last moments when chlorophyll can still be easily detected (day 16, Figure 2D)), and we also see differences in the rate of loss of different photosynthetic proteins in a manner that is not proportional to their normal half-lives (e.g. RBCL is lost rapidly and then PetA and LHCB1 only at a late stage). These changes are not necessary for downsizing the photosynthetic chain alone, and indeed it appears that downsizing can occur when ppGpp signalling is defective (Figure 3A). We therefore think that they are representative of finely regulated changes in the photosynthetic chain that together promote energy dissipation/reduced electron flow and prevent ROS accumulation, and can therefore be considered photoprotective. It is interesting to speculate about why the plant does not remobilise more rapidly to re-activate growth- perhaps it allows a certain regulatory flexibility permitting the plant to liberate some nitrogen in the early stages of starvation without committing to the irreversible destruction of the photosynthetic machinery before it is really necessary.

Also provide discussion in that section about statement line 98 that there is a higher new leaf initiation rate in low ppGpp lines.

We noted the leaf initiation phenotype in the results because it was a striking ppGpp-dependent phenotype. We have added a sentence of interpretation in the Results section. At this point we will not discuss this phenotype in the discussion because we can offer only vague speculation on its cause, and this would risk unfocussing the discussion.

– Page 12, line 300: what about RNA-seq data from the rsh1-1 mutant? It seems an important control that would confirm claim that ppGpp downregulates chloroplast gene expression (page 15, line 348).

We did not obtain RNA-seq data from the *rsh1-1* mutant. However, we do not think that this is essential because previous studies from our team and others support the claim the ppGpp is capable of downregulating chloroplast gene expression in plants:

Sugliani et al. 2016- Quantification of transcription using 4SU RNA labelling

Yamburenko et al. 2015- Nuclear run on using chloroplast extracts.

The studies of Maekawa et al. 2015, Sugliani et al., 2016 and Ono et al., 2020 also show that ppGpp affects the abundance of chloroplast transcripts in Arabidopsis by RT qPCR. Our study extends on these findings by showing that the effect of ppGpp is global, and that it occurs under physiological conditions (i.e. not only during an artificial increase of ppGpp).

­ – Line 320, it is stated that with less ppGpp, nuclear encoded genes for photosynthesis/chloroplast activity are downregulated, yet protein amount are the same, could you discuss turnover rate being possibly slower?

Yes, this is a good point, and we now draw attention to the possibility in the discussion.

Page 15, line 354, “ppGpp acts specifically within the chloroplast” but then how do you explain the lower nuclear encoded genes expression when there is less ppGpp?

Here we mean that the direct mechanistic targets of ppGpp are within the chloroplast. The alteration in nuclear gene expression is likely to be linked to signalling between the chloroplast and the nucleus. Indeed, it is known that inhibition of chloroplast gene expression can activate retrograde signalling (e.g. see Wu et al. 2019, https://doi.org/10.1104/pp.19.00421).

– The overexpressor line “OX:RSH1” is under the 35S promoter which has been shown to possibly yield to silencing (see Grefen et al. Plant Journal, 2010 Vol. 64, pp. 355-365). Have you confirmed the higher expression of RSH1 in your various experiments? A lower expression may explain the higher ppGpp levels than expected shown in Figure 1F and discussed line 453.

The RSH1 overexpression line we use (OX:RSH1 10.4) gives stable and robust overexpression across multiple generations. Seed from the same generation (T4) was used in all experiments reported here, and the RNAseq data also confirm overexpression of RSH1 (Figure 4, source data 1). In addition silencing of OX:RSH1 causes co-suppression of the endogenous *RSH1* gene, and so silenced have an easily identifiable *rsh1* mutant phenotype (like *rsh1-1*). These OX:RSH1 co-suppression lines were reported in Sugliani et al. 2016.

– Page 18, line 403: explain “relaxed”, these bacterial mutants were lacking ppGpp, correct and this work showed that a role for ppGpp inhibiting RNA synthesis? Please provide more explanation.

Yes. In bacteria, the downregulation of rRNA synthesis during amino acid limitation is historically known as the stringent response. Relaxed mutants no longer show activation of the stringent response and do not accumulate ppGpp. The first bacterial RSH enzyme RelA (Relaxed A) was identified and cloned using one of these mutants. We removed the term “relaxed” and added some more explanation.

[Editors’ note: further revisions were suggested prior to acceptance, as described below.]

Editor:The title of the journal should provide a clear indication of the biological system and it should avoid specialist abbreviations and acronyms where possible. Thus, the editor judgeds that use of “ppGpp” needs to be avoided in the title, and ask the authors to change the title such as “A bacterial signaling molecule-mediated brake~”.

We propose the title: “A guanosine tetraphosphate (ppGpp) mediated brake on photosynthesis is required for acclimation to nitrogen limitation in Arabidopsis.”

We think that it is important to keep the word ppGpp in the title because it is a very well-known signalling molecule in bacteria, and is mentioned in thousands of publications (https://www.ncbi.nlm.nih.gov/pmc/?term=ppGpp).

Reviewer 1:The authors have provided a detailed response to my comments and have addressed most of my concerns by revising the manuscript. The authors now note, as a reviewer suggested, that the low Fv/Fm values of the ppGpp over accumulator is due to a high Fo values. Given this, the authors have considerably toned down their argument about ppGpp being an active remodeler of PSII functional assembly, but they still allude to this possibility elsewhere (lines 562-570). I don't protest this hugely but there seems to be a unanimous feeling among the three reviewers that the authors also mention (after line 570) an alternate hypothesis that the decreased PSII activity could simply be a consequence of a coordinated decrease in plastid protein synthesis by ppGpp rather than active regulation of PSII function by ppGpp. A similar downregulation of activity may be apparent for PSI but of course they haven't measured it.

We have added a sentence as suggested to make this point, which we agree with, as clear as possible:

“The most straightforward explanation based on the evidence presented here is that ppGpp acts through a general and coordinated down-regulation of chloroplast gene expression, and further evidence would be needed to determine whether ppGpp is also able to play a more specific role in remodeling photosynthesis. “